# Synthetic cells with self-activating optogenetic proteins communicate with natural cells

Omer Adir [1,2], Mia R. Albalak[1,3], Ravit Abel[1,2], Lucien E. Weiss [4,5,6], Gal Chen [1,3], Amit Gruber [7], Oskar Staufer [8,9,10], Yaniv Kurman[11], Ido Kaminer [11], Jeny Shklover [1], Janna Shainsky-Roitman[1], Ilia Platzman[8,9], Lior Gepstein[7,12], Yoav Shechtman[4,5], Benjamin A. Horwitz [13] & Avi Schroeder [1✉]

Development of regulated cellular processes and signaling methods in synthetic cells is essential for their integration with living materials. Light is an attractive tool to achieve this, but the limited penetration depth into tissue of visible light restricts its usability for in-vivo applications. Here, we describe the design and implementation of bioluminescent intercellular and intracellular signaling mechanisms in synthetic cells, dismissing the need for an external light source. First, we engineer light generating SCs with an optimized lipid membrane and internal composition, to maximize luciferase expression levels and enable high-intensity emission. Next, we show these cells' capacity to trigger bioprocesses in natural cells by initiating asexual sporulation of dark-grown mycelial cells of the fungus *Trichoderma atroviride*. Finally, we demonstrate regulated transcription and membrane recruitment in synthetic cells using bioluminescent intracellular signaling with self-activating fusion proteins. These functionalities pave the way for deploying synthetic cells as embeddable microscale light sources that are capable of controlling engineered processes inside tissues.

[1] The Louis Family Laboratory for Targeted Drug Delivery and Personalized Medicine Technologies, Department of Chemical Engineering, Technion, Haifa, Israel. [2] The Norman Seiden Multidisciplinary Program for Nanoscience and Nanotechnology, Technion, Haifa, Israel. [3] The Interdisciplinary Program for Biotechnology, Technion, Haifa, Israel. [4] Department of Biomedical Engineering, Technion, Haifa, Israel. [5] Lorry I. Lokey Interdisciplinary Centre for Life Sciences and Engineering, Technion, Haifa, Israel. [6] Department of Engineering Physics, Polytechnique Montréal, Montréal, ON, Canada. [7] Sohnis Research Laboratory for Cardiac Electrophysiology and Regenerative Medicine, the Rappaport Faculty of Medicine and Research Institute, Technion, Haifa, Israel. [8] Department for Cellular Biophysics, Max Planck Institute for Medical Research, Heidelberg, Germany. [9] Institute for Molecular Systems Engineering (IMSE), Heidelberg University, Heidelberg, Germany. [10] Max Planck School Matter to Life, Heidelberg, Germany. [11] The Andrew and Erna Viterbi Faculty of Electrical and Computer Engineering, Technion, Haifa, Israel. [12] Cardiology Department, Rambam Health Care Campus, Haifa, Israel. [13] Faculty of Biology, Technion - Israel Institute of Technology, Haifa, Israel. ✉email: avids@technion.ac.il

In recent years, there has been a growing interest in the field of synthetic cells (SCs) for therapeutics, diagnostics and research on the origin of life. These cell-mimicking microparticles are designed from the bottom-up to reconstitute various processes of living cells and introduce entirely new functionalities that are not present naturally[1–3]. Biological processes such as protein expression, ATP production, DNA replication and cytoskeleton re-arrangement have all been successfully reconstructed in SCs, yielding new insights in the isolated and controlled model environment[4–7]. Due to their tunable properties (i.e., size, composition, membrane rigidity) and engineerability, promising clinical applications in diagnostics and therapeutics are being realized as well. These include insulin-secreting synthetic beta cells and therapeutic-protein-producing SCs that express Pseudomonas exotoxin A (PE) to eliminate 4T1 breast cancer cells, both of which have already been tested in animal models in-vivo[8,9]. The implementation of increasingly complex functions in SC technologies requires inputs and outputs, i.e. cell signaling, that can be used orthogonally[10,11]. This is of particular importance for interfacing SCs in living tissue.

Previous reports have demonstrated chemical communication between SCs and bacterial cells as well as between SCs[12,13]. For example, isopropyl β-D-1-thiogalactopyranoside (IPTG), arabinose and C6-HSL (N-(3-oxohexanoyl)-L-homoserine lactone) have been used as signaling molecules to control gene expression and differentiation in SCs[14,15]. Light has also been applied to stimulate cellular processes in SCs. UV radiation (365 nm) was utilized to initiate transcription and activate communication in SCs and tissues by unlocking DNA photo-caging[16,17]. In another example, SCs with bacteriorhodopsin incorporated in their membranes formed a proton gradient and could drive ATP synthase activity upon illumination with blue light[18]. This wide diversity of light-responsive elements, activated by light from different parts of the spectrum, provides multichannel signaling opportunities. Problematically, using light over much of the visible spectrum for in-vivo applications often requires invasive transplantations of external light sources[19,20]. A translational alternative for using external light sources is to generate light from SCs localized in the target area. This can be achieved by exploiting bioluminescent reactions catalyzed by enzymes from the luciferase family, which have been widely used in research for biological reporter assays and in-vivo imaging[21,22].

The use of luciferase-generated bioluminescence for activation of optogenetic proteins, genetically encoded light responsive elements that can modulate cellular states, has been explored in diverse cellular processes, including ion channel activation for neuromodulation and transcriptional control in eukaryotic cells[23,24]. Due to the weak intensity of luciferase in comparison to external light sources (lasers and LEDs), many of these applications focused on intracellular cascades, in which the luciferase and the target photo-responsive protein are in close proximity. To maximize the efficiency of activation, fusion proteins of luciferase and the target light responsive protein were designed, harnessing bioluminescence resonance energy transfer (BRET)[25,26]. Recently, bioluminescent trans-synaptic activation between physically connected neurons was enabled by accumulation of luciferase proteins secreted by pre-synaptic cells into the synaptic space, yielding a local concentration that was sufficient to activate a light sensitive protein in the post-synaptic cells[27].

Here, we harness bioluminescence to engineer intercellular and intracellular signaling mechanisms in SCs for the purpose of activating cellular processes in both natural and SCs. The SCs were composed of giant unilamellar vesicles (GUVs) encapsulating a bacterial-based cell-free protein synthesis (CFPS) system (Fig. 1a). To enable intercellular signaling between a SC and a natural cell, light-generating SCs were designed to express high levels of Gaussia luciferase (Gluc) in order to enhance their light emission intensity. These cells were then used to activate photoconidiation in fungal cells, demonstrating their capacity to control a biological process in a subsequent population of natural cells. Next, intracellular bioluminescent signaling processes were engineered in SCs. In order to utilize light-responsive proteins that required higher intensities, self-activating fusion proteins were engineered by coupling Gluc with photo-responsive proteins, facilitating their activation by BRET. This approach was used to control transcription in SCs using a bioluminescent fusion protein of Gluc and the light-activated transcription factor EL222. Light-controlled activation of membrane recruitment in SCs was achieved as well, with an additional BRET-based signaling mechanism utilizing a fusion protein of Gluc and iLID, that dimerized with a sspB-tagged protein when the bioluminescent reaction was initiated. Altogether, these SC signaling functionalities present opportunities for bioluminescent activation and control of synthetic and natural cells alike.

## Results

**Optimizing the lipid composition of light-interacting SCs**. The phospholipid membrane is the main physical barrier for light emission or light absorption in SCs composed of GUVs, and must therefore have favorable optical properties. Hence, the first step in constructing light-generating and light-responding SCs is to optimize their membrane composition. In this work, we focused on interactions with blue light (480 nm), which can be used to activate photoreceptors such as light-oxygen-voltage-sensing (LOV) domains present in many photoactivatable proteins, and is emitted by certain types of luciferases, including Renilla and Gaussia luciferase[28,29]. Furthermore, the use of blue light for in-vivo applications traditionally poses a challenge for non-implanted sources due to its poor tissue penetration.

Factors affecting light transmission through the lipid membrane were investigated and the phospholipid composition of the membrane was altered accordingly to minimize attenuation. First, the absorbance of 480 nm light by 100-nm liposomes composed of a single phospholipid type was measured (Fig. 1b). This liposome size was selected for this measurement due to its lower polydispersity index (Fig. S1), thereby reducing sample-to-sample variation originating from size-dependent light interactions (i.e. light scattering). The lipid-light interactions demonstrated a positive correlation between the phospholipid tail length and light absorbance (Fig. 1b). L-α-phosphatidylcholine, hydrogenated (Soy), (HSPC) which is composed of 88.6% of 18:0 fatty acids had the highest absorbance, which was more than 1.5-times higher than 1,2-dipalmitoyl-sn-glycero-3-phosphocholine (DPPC) with 16:0 tails and 2.6-times higher than 1,2-dimyristoyl-sn-glycero-3-phosphocholine (DMPC) with 14:0 tails. This trend was also apparent in liposomes composed of unsaturated phospholipids, in which the 18:1 lipid 1,2-dioleoyl-sn-glycero-3-phosphocholine (DOPC) had higher light attenuation compared to 16:0,18:1 lipid 1-palmitoyl-2-oleoyl-glycero-3-phosphocholine (POPC). Moreover, liposomes composed of unsaturated lipids had lower absorbance relative to those made of saturated-lipids with similar tail lengths (more than 1.7-times difference between the absorbance of HSPC and DOPC).

While both DMPC and POPC liposomes had low light absorbance, the latter was selected as the main lipid in the SC due to its lower melting temperature ($T_m$) of $-2\,^{\circ}$C which enabled it to remain in a liquid phase during the SC production process performed at 4 °C. Although cholesterol does contribute to the light absorbance (Fig. S2), its presence in the SC membrane (1:1 w/w ratio) was crucial to enhance membrane stability, which is important since the incubation during protein production is performed at 37 °C.

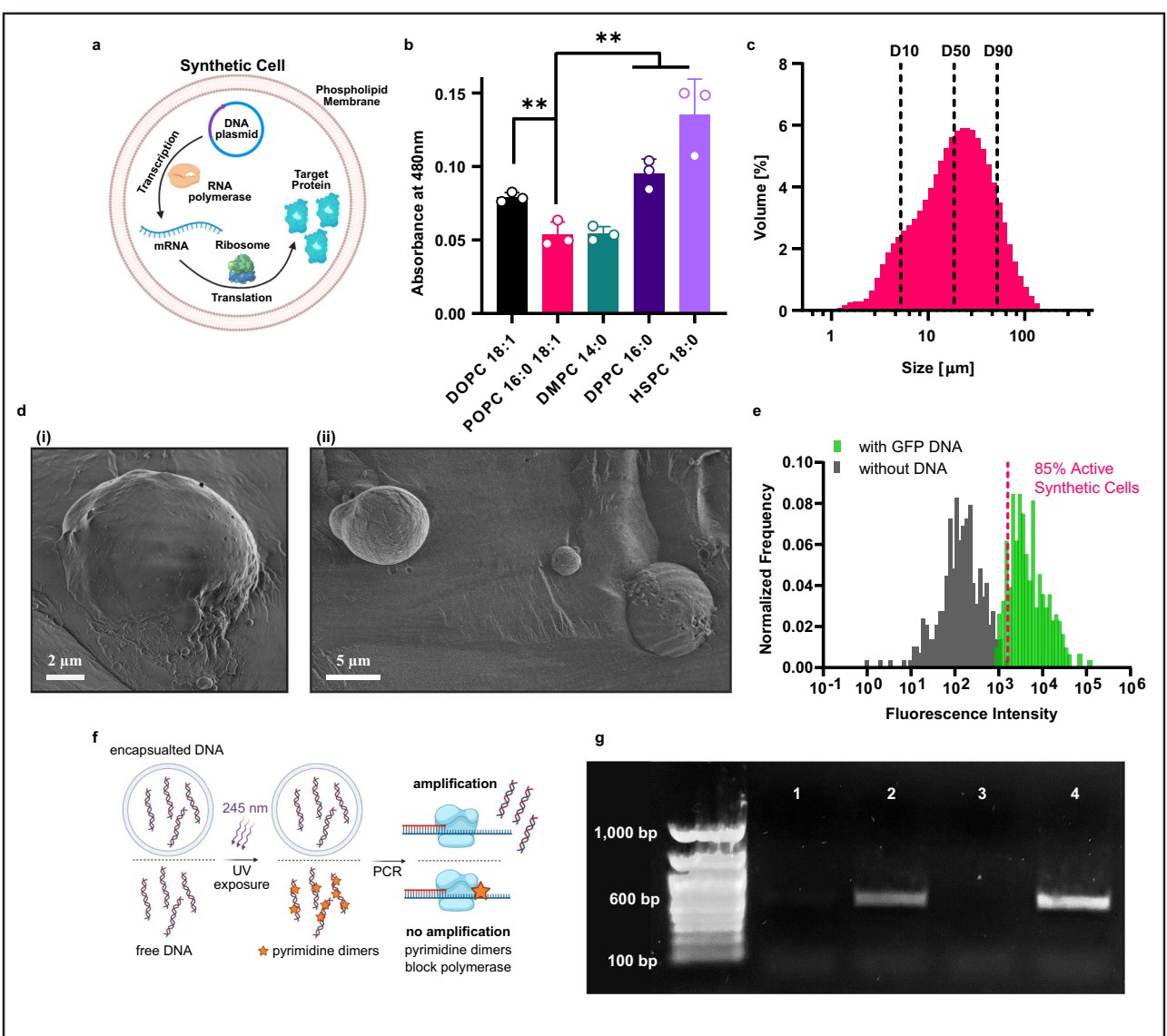

**Fig. 1 Optimizing the lipid composition for light-interacting synthetic cells. a** Illustration of a protein producing synthetic cell. **b** The effect of the lipid membrane composition on blue light absorbance in liposomes. Data is expressed as the mean ± standard deviation ($n = 3$ independent samples). Nested two-tailed $t$-test $P$ values; **$p = 0.0079$, **$p = 0.0058$, **$p = 0.0055$ for comparisons of POPC vs. DOPC, DPPC and HSPC, respectively. **c** Size distribution of synthetic cells with 1:1 (w/w) POPC:cholesterol membrane composition. Data is expressed as mean of $n = 3$ independent samples. **d** Morphology of 1:1 (w/w) POPC:cholesterol synthetic cells imaged with cryogenic scanning electron microscopy (cryo-SEM). **e** The percentage of active synthetic cells measured using imaging flow cytometry of GFP-expressing synthetic cells ($n = 331$ synthetic cells without DNA and $n = 308$ synthetic cells with GFP DNA). Frequency is normalized to the total number of cells in each sample. **f** Illustration of a linear DNA oligonucleotide, encapsulated in a synthetic cell or free in solution, exposed to UV radiation. Formation of pyrimidine dimers is detected using PCR amplification. **g** Gel electrophoresis of the amplified PCR product of linear DNA after exposure to UV radiation with and without encapsulation in synthetic cells. Lane 1: free DNA exposed to UV. Lane 2: encapsulated DNA exposed to UV. Lane 3: no DNA control. Lane 4: untreated DNA control. This experiment was reproduced n = 3 times. Source data are provided as a Source Data file.

Once the lipid composition of the membrane was selected, SCs were produced using the emulsion transfer method. This method's simplicity, high encapsulation efficiency, and high yield make it preferable for achieving high overall protein production[30]. Light scattering was used to determine the obtained size distribution, spanning between 1-200 μm with average D10, D50 and D90 values (corresponding to the 10%, 50% and 90% marks in the cumulative size distribution curve) of 5.2, 18.6 and 51.8 μm, respectively (Fig. 1c). Cell morphology and size range were further validated with cryogenic scanning electron microscopy (cryo-SEM, Fig. 1d, S3). Imaging flow cytometry was then used to assess the percentage of active SCs (executing the

transcription and translation processes), by measuring the fluorescence intensity of SCs encapsulating a CFPS solution and GFP encoding DNA (Fig. 1e). 85% of the SCs were active and displayed higher fluorescence levels than the maximal background auto-fluorescent signal that was measured using a population of SCs without a DNA template.

Next, we scanned the absorbance spectrum of the tested phospholipids at different wavelengths (230–800 nm) to investigate whether the absorbance correlations recognized for 480 nm light were maintained for other wavelengths that could also be used for light signaling (Fig. S4). The obtained absorbance data of each phospholipid was fitted with a spectral line shape Lorentzian

function ($R^2 \geq 0.9993$) to extrapolate its absorbance properties in shorter wavelengths (down to 120 nm). From the measured data and Lorentzian function, we determined that the correlation between the phospholipids' tail length and absorbance continued in other parts of the spectrum. Interestingly, unsaturated phospholipids that had lower absorbance at 480 nm than phospholipids with saturated tails of the same length, demonstrated higher peak absorbance in the range of 160–200 nm according to their Lorentzian function (characteristic of carbon double bonds[31]). Furthermore, all of the liposome formulations exhibited a similar absorbance trend with absorption resonance wavelengths in the UVC range (200–290 nm), and little light attenuation at wavelengths longer than 480 nm (Fig. S4). The high agreement between the measured data and fitted Lorentzian functions indicates that light scattering in these measurements was negligible.

Considering the wide application range of SCs as model systems for various cellular processes, and as diagnostic and therapeutic tools, we further investigated the interaction of the SCs' phospholipid membrane with light, and specifically its implications on the SCs' DNA integrity. We hypothesized that the phospholipid membrane's high UV absorption could protect DNA from UV radiation damage and tested this using our SC platform (Fig. 1f, g).

Exposure of DNA to UV radiation is known to lead to the formation of pyrimidine dimers and other DNA photoproducts that damage the DNA functionality and lead to subsequent biological effects (Fig. 1f)[32,33]. We exposed a 600-bp linear DNA oligonucleotide to UV radiation for 20 min before or after its encapsulation in a SC and detected the formation of DNA lesions with the polymerase chain-reaction (PCR), which is inhibited by these lesions[34]. Encapsulation of the DNA in SC yielded evident PCR amplification indicating improved UV protection in comparison to the DNA that was irradiated prior to its encapsulation and yielded almost no PCR product at all (Fig. 1g). In addition to their eminent role as compartmentalizing structures, this highlights the importance of phospholipid membranes as UV-protecting scaffolds.

**Engineering light-generating SCs.** Following the optimization of the SCs' membrane, we engineered SCs capable of generating light by expressing luciferase for catalyzing a photon-emitting enzymatic reaction. Considering energy consumption, emitted wavelength, and total light emission intensity parameters, we chose to focus on two luciferase types, *Renilla* luciferase (Rluc) and a mutated variant of *Gaussia* luciferase (M43L, M110L, and lacking the secretion signal peptide sequence, amino acids 2–17)[35]. The luciferases were sourced from the organisms *Renilla reniformis* and *Gaussia princeps* respectively. Both types of luciferase catalyze an ATP-independent bioluminescent oxidation reaction of the substrate coelenterazine (CTZ), conserving energy for the SC[29,36]. In terms of emission wavelength, both enzymes generate photons in the blue-range, suitable for LOV domain activation. Nevertheless, while Rluc has rapid flash kinetics, the Gluc variant exhibits brighter and longer half-life of illumination (glow-like kinetics)[35].

Initial comparison of light production of both luciferase types in CFPS reactions with *E. coli* BL21(DE3) lysates indicated 2500-fold higher light emission in Rluc-expressing SCs in comparison to weak illumination produced from the Gluc-expressing SCs (Fig. 2a). Most likely, the poor production of Gluc in the SCs was due to misfolding of its five disulfide bonds in the reducing environment of the SCs that contains both 1,4-Dithiothreitol (DTT) and 2-mercaptoethanol[37]. Moreover, oxidative folding for

disulfide bond generation in *E. coli* is performed in the periplasm and not in the reducing environment of the bacterial cytoplasm which is a main component of the lysate[38]. Therefore, the SCs' internal composition was redesigned. Reducing agents DTT and 2-mercaptoethanol were excluded from the internal solution, and glutathione and disulfide bond isomerase C (DsbC) from *E. coli* were added to improve Gluc folding[39]. A range of oxidized and reduced glutathione concentrations were added to the inner solution of the SCs and light emission after CTZ addition was measured (Fig. S5). 4 mM of oxidized and 1 mM of reduced glutathione were found to be optimal for Gluc production with the *BL21(DE3)* lysate. After further addition of 75 μg ml$^{-1}$ of DsbC (Fig. S6) to the glutathione-supplemented SC internal solution, light emission from Gluc-expressing SCs was more than 100-fold higher than that of the Rluc-expressing SCs (Fig. 2b), and was easily visible by eye in solution (Fig. 2c). Furthermore, these engineered SCs generated a 10-fold great photon emission compared to SCs expressing Gluc using a commercial CFPS reaction (Fig. S7). In contrast, the commercial CFPS solution without encapsulation in SCs outperformed our self-prepared CFPS solution. This difference highlights the importance of optimizing the SCs' internal solution properties (such as density and osmolality) to fit the encapsulation process. Gluc expression in SCs was quantified by western blot analysis demonstrating production of 40.8 ± 4.0 ng μl$^{-1}$ (Fig. 2d) of protein. This translates to an average of 24 pg of Gluc per SC, considering a concentration of $1.68 \times 10^6$ SCs ml$^{-1}$, as quantified using multispectral imaging flow cytometry.

The protein production and light-emission properties of Gluc-expressing SCs were further characterized under different incubation times, temperatures (Fig. S8), SC and substrate concentrations. Protein-production kinetics in SCs at 37 °C were monitored using light-emission assays performed at different incubation times (Fig. 2e). Gluc levels increased over the first 60 min and plateaued at 90 min. Light emission kinetic measurements demonstrated a $t_{1/2}$ of two minutes for SCs incubated with 100 μM of CTZ (Fig. 2f). After 15 min, the illumination intensity was approximately 7% of the initial intensity measured and still more than 100,000 fold higher than the intensity measured from control SCs without a DNA template encoding for luciferase.

Next, the dependency of light intensity on the SC concentration was demonstrated (Fig. 2g). A concentration of 420,000 SCs ml$^{-1}$ was found to generate the maximal light intensity upon CTZ addition and was used for the subsequent experiments. In comparison, 2 and 4-times higher cell concentrations and 2.5 and 5-times lower cell concentrations had significantly lower light emission. Increasing concentrations of SCs increases the total Gluc concentration, but also the total phospholipid and cholesterol concentrations which attenuate light emission from the solution. The yield of emitted light is thus balanced between these two parameters.

Light intensity was also controlled by altering the substrate concentration, demonstrating an increase in the generated signal between 10 μM and 100 μM CTZ, and a slight decrease for 200 μM CTZ (Fig. 2h). Lower light intensities were also detectable using lower concentrations of substrate, down to 10 nM of CTZ (Fig. S9). Temporal control of illumination was achieved by timed addition of the substrate to the solution (0.125 nmol CTZ). The emission of SCs was restored when an additional dose of 0.125 nmol of substrate was added. The light intensity generated upon additional doses of substrate following the decay of the initial light emission reached intensities similar to those produced in the first illumination pulse (Fig. 2i). This regeneration ability enables high light intensity levels to be maintained when prolonged illumination is required, and to perform multiple activations at different time points.

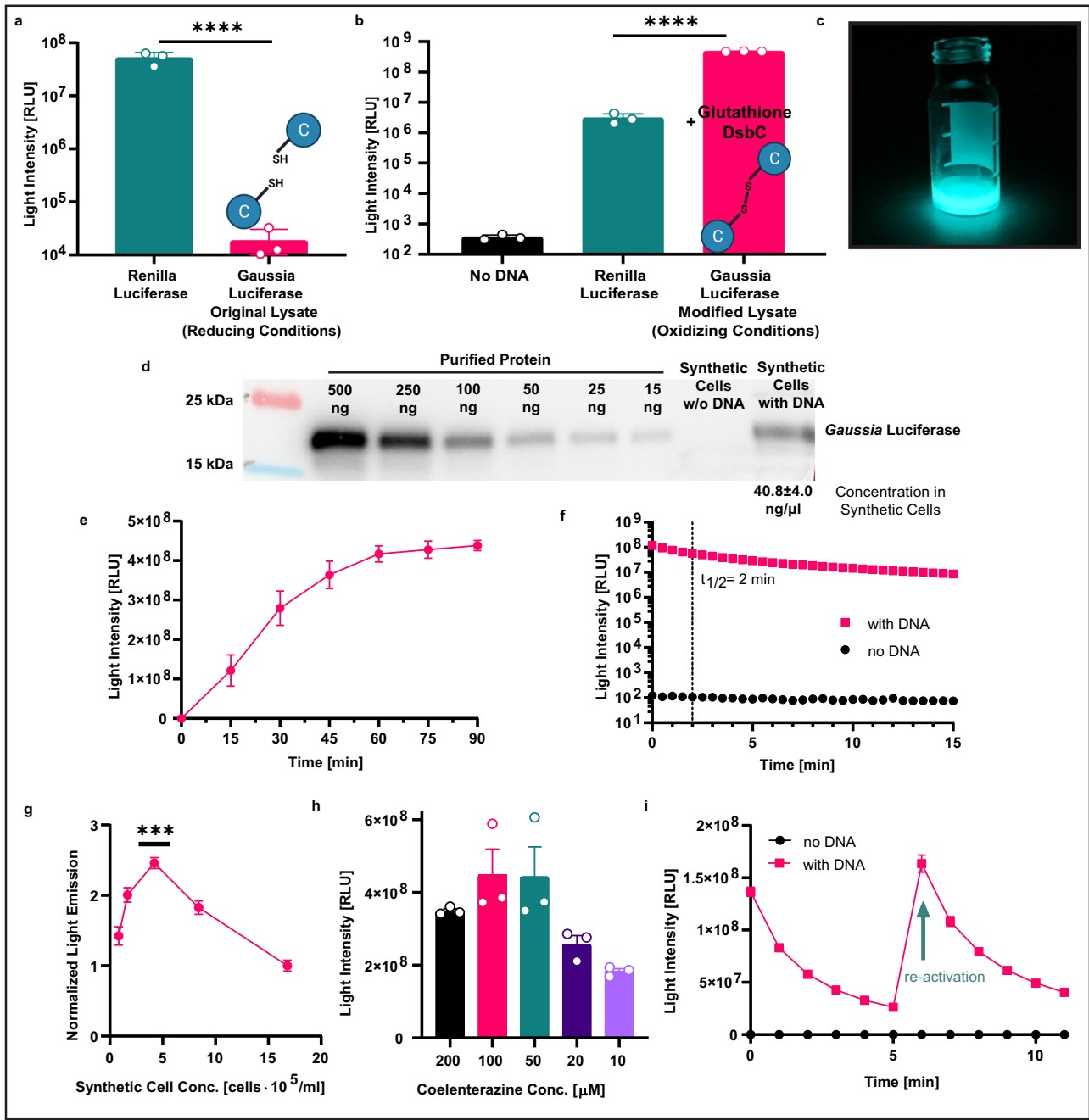

**Fig. 2 Engineering light-generating synthetic cells. a** Comparison of light emission in cell-free protein synthesis reactions expressing *Renilla* Luciferase (Rluc) or *Gaussia* luciferase (Gluc) using unmodified lysate (with reducing conditions). Nested two-tailed *t*-test *P* value; ****$p = 3.82 \times 10^{-6}$. **b** Light emission by Gluc-expressing synthetic cells containing lysate modified with glutathione and disulfide bond isomerase C (DsbC) to produce an oxidizing environment, compared to light emission by Rluc-expressing synthetic cells. Nested one-way ANOVA with multiple comparisons test adjusted *P* value; ****$p = 6.6 \times 10^{-14}$. **c** Light emission from a Gluc-expressing synthetic cell solution. **d** Western blot quantification of Gluc production in synthetic cells. *Gaussia* luciferase concentration in synthetic cells is calculated based on $n = 2$ independent experiments. **e** Gluc production kinetics in synthetic cells at 37 °C. **f** Kinetics of the Gluc enzymatic reaction in synthetic cells after one addition of 100 μM coelenterazine. **g** Light emission from Gluc-expressing synthetic cells diluted to different concentrations after incubation. Nested one-way ANOVA with multiple comparisons test adjusted *P*-values; ****$p = 8.3 \times 10^{-7}$, ***$p = 0.00062$, ****$p = 3.7 \times 10^{-5}$, ****$p = 1.3 \times 10^{-8}$, for comparisons of $4.2 \times 10^5$ synthetic cells ml$^{-1}$ to $0.8 \times 10^5$, $1.7 \times 10^5$, $8.4 \times 10^5$, and $16.8 \times 10^5$ synthetic cells ml$^{-1}$, respectively. **h** Light emission in ranging coelenterazine concentrations by Gluc-expressing synthetic cells. Data is expressed as a mean ± s.e.m. ($n = 3$ independent samples). **i** Temporal control over light emission in Gluc-expressing synthetic cells with two timed additions of 0.125 nmol coelenterazine (second addition marked with a green arrow). for **a**, **b**, **e**, **f**, **g**, **i**. Data is expressed as a mean ± standard deviation ($n = 3$ independent samples). Source data are provided as a Source Data file.

**Light-producing SCs activate fungal cells**. To demonstrate intercellular signaling between light-producing SCs and light-responsive natural cells we employed the soil fungus *Trichoderma atroviride*. Two blue-light regulator proteins (BLR-1, a LOV-domain protein, and BLR-2) control the photo-activation of this fungus in response to blue light, triggering several downstream processes including conidiation and the expression of the DNA repairing photolyase enzyme[40–42]. The level of fungal conidiation depends on the total light exposure with no dependency on the light intensity or duration (abides the Bunsen-Roscoe law of reciprocity), and thus fungal cells can be activated with continuous low-intensity light emission[40]. *T. atroviride* were plated and grown for 48 h in dark room conditions to prevent activation by external light. 24 h after a 1-minute exposure to blue LEDs (15 mW cm$^{-2}$), a peripheral ring of spores was evident on the border of the fungi colony (Fig. S10). Alternatively, SC illumination was supplied in two consecutive induction rounds, each with 500 μL of SCs supplemented with 100 μM CTZ that were added to a restricted section in the fungi plate and incubated for 15 min (Fig. 3a). After an incubation period of 24 h in the dark following induction, conidiation was quantified by calculating the percentage of sporulated area out of the total area exposed to SCs (Fig. 3b, c). Fungi incubated with Gluc-expressing SCs demonstrated an average of 9.1% sporulated area, a 23-times increase in comparison to fungi incubated with SCs without a DNA template (Fig. 3c).

The activation of the BLR pathway was dependent on the SC density. A range of SC dilutions (between 4200 to 420,000 cells ml$^{-1}$) were incubated on the *T. atroviride* plates and conidiation was quantified (Fig. 3d). A minimal concentration of 84,000 SCs ml$^{-1}$ was required for activation of conidiation, with a ~2.2-times increase in sporulated area observed between 84,000 cells ml$^{-1}$ and 420,000 cells ml$^{-1}$. In order to estimate the conidiation activation levels of the fungal colonies more precisely, a light dose calibration curve ranging from 10 μE m$^{-2}$ to 22,500 μE m$^{-2}$ was generated using a blue LED lamp (Fig. 3e, black circles). The sporulation level in each treatment was measured by suspending the spores from the colonies in water and measuring the scatter at 600 nm. A Michaelis–Menten model was used to fit the measured data and to calculate the equivalent light dose and activation level of the SC treatment. The Michaelis–Menten Eq. (1):

$$OD_{600} = (V\max \cdot Total\ Light\ Dose)/(Km + Total\ Light\ Dose) \quad (1)$$

provided best fit values of 0.4157 (absorbance units at 600 nm) for Vmax (denoting the optical density at saturation) and 2407 μE m$^{-2}$ for Km (denoting the light dose required to reach 50% of the saturation level) with $R^2 = 0.9582$. The calculated light dose for SCs was approximately 4552 μE m$^{-2}$ (Fig. 3e, pink square), which is higher but in the 95% confidence interval (CI) range of the Km of the curve (95% CI of 1186 to 5253 μE m$^{-2}$). Based on the best fit Vmax value of the curve, the activation level of the SC treatment was calculated to be approximately 65% of the saturation level.

The dependency of *T. atroviride*'s photo-activation on the SC concentration resembles the cell density dependency that is a key characteristic of quorum sensing mechanisms in different species[43,44]. Nevertheless, natural quorum sensing mechanisms differ from the signaling mechanism demonstrated here as they are based on chemical entities (and not light) and contain a positive feedback loop to initiate a population coordinated response. Moreover, the use of light as a signaling module in this mechanism has the unique property of enabling signal transmission even when the cell populations are separated by a physical transparent barrier.

*Bioluminescent self-activation of transcription in SCs*. Next, we demonstrate auto-activation of two different intracellular

processes in SCs using bioluminescent reactions: induction of protein expression and membrane recruitment. To achieve this, we designed self-activating fusion proteins composed of Gluc and a light-responding domain that initiate these processes.

Control over protein expression was performed using the light-inducible bacterial transcription factor EL222, that dimerizes when exposed to blue light and binds to a specific region in the pBLind promoter to initiate transcription (Fig. 4a)[45,46]. To simplify the integration of EL222 into the SC inner solution, calibration of the required EL222 concentration was initially performed in CFPS reactions (before encapsulation in SCs) using an external LED blue light source for activation. Rluc was used as a reporter protein, and a DNA plasmid containing its reading frame after the pBLind promoter was prepared. Purified EL222 was added in concentrations of 2.5 μM, 5 μM and 10 μM to the CFPS mix and incubated for 1 h in dark or light conditions (Fig. 4b). Rluc expression levels were analyzed by adding CTZ and measuring luminescence. EL222 had the most significant light-to-dark ratio at 2.5 μM, and demonstrated significant light-to-dark differences at 5 and 10 μM. Production of monomeric RFP (mRFP1), that is characterized by a longer folding time, in CFPS reactions with 2.5 μM EL222 was subsequently tested (Fig. 4c). Elevated mRFP1 levels were evident in the reaction mix containing EL222 and mRPF1 DNA after 5 h of incubation with blue light, in comparison to the same solution incubated in dark conditions. The fluorescence levels continued to increase for 12 h (Fig. 4c, pink and black filled circles). Some increase in signal was evident in the dark-incubated sample without DNA (Fig. 4c, hollow black circles), and is associated with increase in auto fluorescence over time.

After establishing the reaction conditions in CFPS, we integrated the EL222 system in SCs and activated expression of Rluc using an external LED light. The higher light absorbance of SCs compared to CFPS required increasing the light intensity used for activation, while avoiding overheating that might lead to protein denaturation. Therefore, the light intensity was raised from approximately 12 W m$^{-2}$ to 19 W m$^{-2}$ with on-off intervals of 20 and 70 s, respectively. Under these conditions, SCs encapsulating EL222 monomers and a DNA plasmid expressing Rluc under the pBLind promoter showed 2.4-fold increase in Rluc expression in light vs. dark conditions (Fig. 4d). In comparison, SCs with EL222 and no DNA or a DNA plasmid with Rluc expressed under the viral T7 promoter which is not specific to EL222, showed negligible changes.

In order to activate EL222 with bioluminescence, close proximity between the light source and the responsive protein is required. For this purpose, we designed a fusion protein, with Gluc on its N-terminal end connected through a flexible peptide linker to EL222 on its C-terminal end (Fig. S11). This construct utilizes BRET to exploit the energy from the bioluminescent luciferase reaction, to directly activate the target protein (Fig. 4e). We added the Gluc-EL222 protein to the SC interior instead of the native EL222, and followed mRFP1 production with and without the addition of CTZ (Fig. 4f, S12). A 3.2-fold increase in mRFP1 production was observed when CTZ was added to the SCs compared to SCs that were incubated in the dark without CTZ. This demonstrates the functionality of the fusion protein as a self-activating transcription factor in SCs.

**Bioluminescent self-activated membrane recruitment in SCs**. To further explore the potential of bioluminescent self-activating processes in SCs, we incorporated light-controlled protein recruitment capabilities to the SC membrane. Specifically, this was performed using a hetero-dimerization reaction with one monomer conjugated to the SC membrane (iLID) and the other

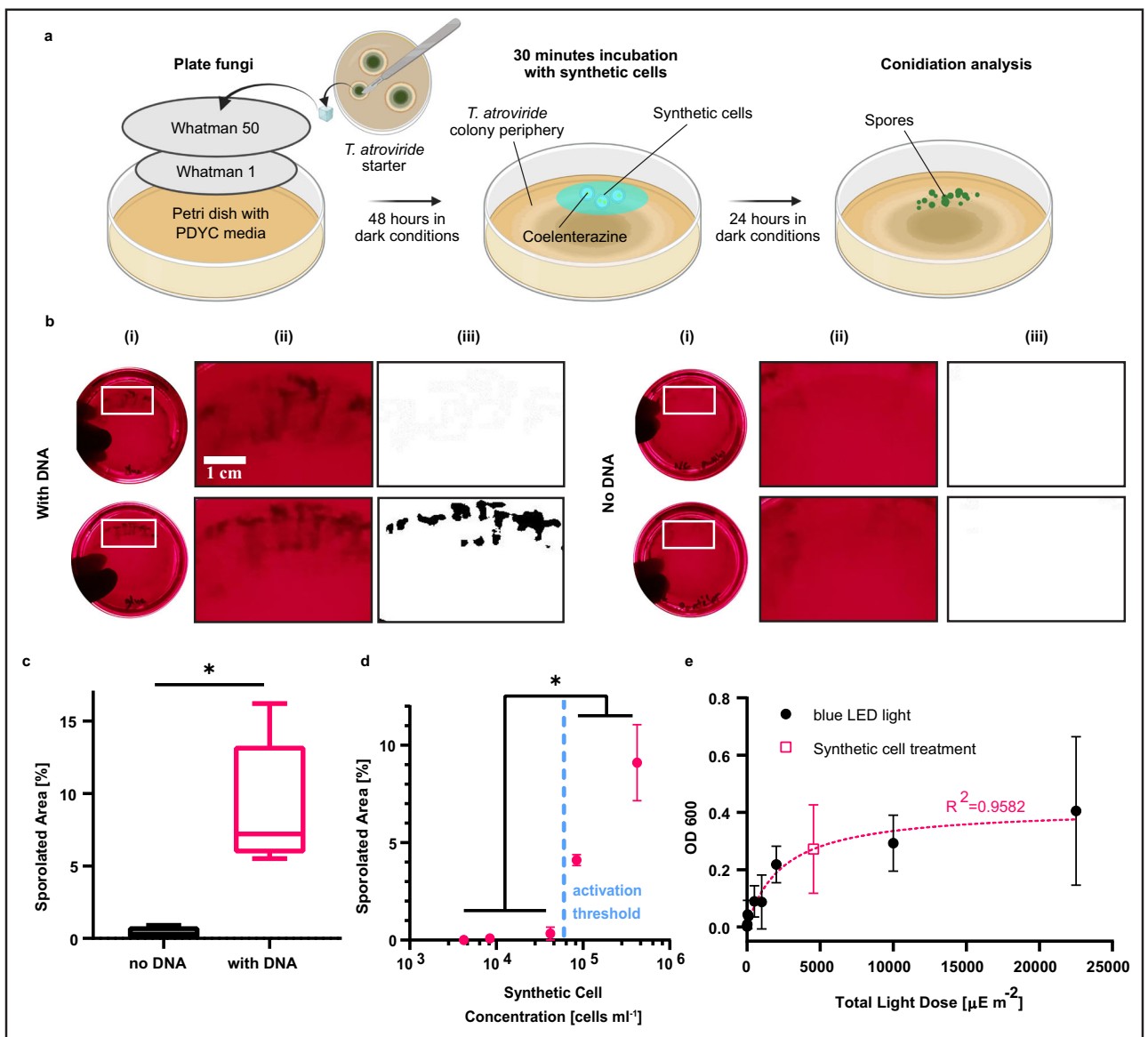

**Fig. 3 Activation of fungal cells using light-producing synthetic cells. a** Illustration of the experimental setup for photo-activation of conidiation in *Trichoderma atroviride* with *Gaussia* luciferase (Gluc)-expressing synthetic cells. **b** (i) Representative images of *Trichoderma atroviride* plates after exposure to Gluc-expressing synthetic cells or synthetic cells without DNA. (ii) A magnified image and a (iii) black and white thresholded image of the plate area marked with a white rectangle in which the synthetic cells were localized. **c** Quantitative analysis of the sporulated area out of the total area exposed to synthetic cells. Horizontal lines indicate median values; boxes indicate quartiles 1 and 3; whiskers indicate the min and max values ($n = 5$ independent samples). Welch's two-tailed *t*-test P value; *$p = 0.0103$. **d** The effect of varying synthetic cell concentrations on photo-activation of conidiation in *Trichoderma atroviride*. Data is expressed as a mean ± s.e.m. ($n = 5$ independent samples for 420,000 synthetic cells ml$^{-1}$, $n = 3$ independent samples for all other synthetic cell concentrations). Welch's two-tailed *t*-test P-values; *$p = 0.01$, ***$p = 0.0009$, **$p = 0.0098$, **$p = 0.0026$, **$p = 0.0095$, **$p = 0.0045$, for comparisons of 420,000 and 84,000 synthetic cells ml$^{-1}$ vs. 42,000, 8400 and 4200 synthetic cells ml$^{-1}$, respectively. **e** A light dose-response calibration curve was generated in *Trichoderma atroviride* using a blue LED (black circles). A Michaelis-Menten curve was fitted to the data (dotted pink line, $R^2 = 0.9582$) and the total light dose of the synthetic cell treatment was calculated according to the fitted model (pink square). Data is expressed as mean ± standard deviation ($n = 2$ independent samples for light doses of 20, 1000, 2000 µE m$^{-2}$ and the synthetic cells samples, $n = 3$ independent samples for all other light doses). Source data are provided as a Source Data file.

free in solution (sspB-Nano, Fig. S13)[47]. We incorporated DGS-NTA-Ni lipids into the SC membrane to bind his-tagged iLID and monitored the localization of mRFP1 fused to the sspB-Nano protein. In these experiments, we prepared the SCs with a droplet-based manual emulsification method, which is based on charge-mediated liposome fusion inside surfactant-stabilized droplets. This method enables higher flexibility in lipid selection in comparison to the emulsion-transfer method, which allowed us to fabricate SCs with a membrane composition of

73.5 mol% POPC, 20% DOPG, 5% DGS-NTA-Ni, 1% DOPE-PEG4-biotin and 0.5% DOPE-Cy5[48].

Gluc-expressing CFPS reactions did not provide sufficient light to activate the LOV domain of the iLID and initiate sspB-Nano recruitment to the SC membrane (Fig. S14). We therefore engineered a second fusion protein, N-terminal Gluc fused to C-terminal iLID with a linker peptide and a N-terminal his-tag, harnessing BRET again for efficient activation of the light-responsive domain (Fig. 5a, S15). This structure facilitates

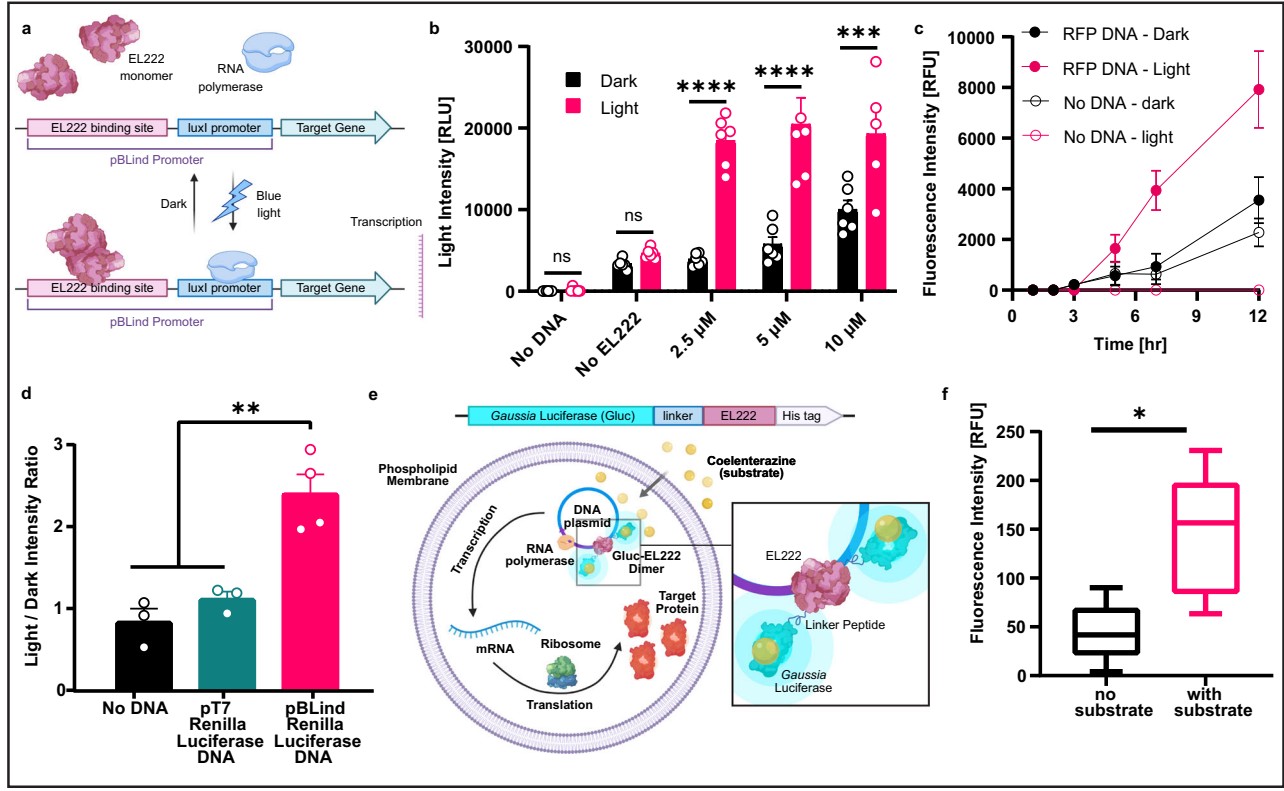

**Fig. 4 Bioluminescent signaling self-activates transcription in synthetic cells. a**, Illustration of the light-dependent transcription mechanism mediated by the transcription factor EL222. **b**, EL222 concentration affected the production of *Renilla* luciferase (Rluc) in cell-free protein synthesis (CFPS) reactions under light and dark conditions. Data is expressed as a mean ± standard deviation ($n = 5$ independent samples for 10 µM EL222 under light conditions, n=6 independent samples for all other experimental groups). Two-way ANOVA with multiple comparisons test adjusted *P*-values; ****$p = 3.26×10^{-8}$, ****$p = 2.51×10^{-8}$, ***$p = 0.0005$, for comparison of light vs. dark conditions of 2.5, 5 and 10 µM, respectively. **c** RFP production kinetics in CFPS reactions supplemented with EL222 under dark and light conditions, with or without RFP DNA. Data is expressed as a mean ± s.e.m. ($n = 4$ independent samples for the "no DNA" groups, $n = 6$ independent samples for other groups). **d** Light-to-dark ratio of Rluc expression in synthetic cells containing EL222 and a DNA plasmid expressing Rluc under different promoters. Data is expressed as a mean ± s.e.m. ($n = 4$ independent samples for the pBLind promoter group, $n = 3$ for other experimental groups). One-way ANOVA with multiple comparisons test adjusted *P* value; **$p = 0.0016$, **$p = 0.005$, for comparisons of the pBLind group vs. the no DNA and pT7 groups, respectively. **e** A block diagram of the Gluc-EL222 fusion protein elements. Below, a schematic representation of a synthetic cells containing the Gluc-EL222 fusion protein for bioluminescent activation of transcription. **f** RFP production in synthetic cells containing the Gluc-EL222 protein with or without addition of coelenterazine. Horizontal lines indicate median values; boxes indicate quartiles 1 and 3; whiskers indicate the min and max values ($n = 5$ independent samples). Student's two-tailed *t*-test *P* value; *$p = 0.0134$. Source data are provided as a Source Data file.

binding of the fusion protein to the SC membrane from the Gluc end and exposes the iLID to the extracellular environment with lower steric interference (Fig. 5a). Gluc-iLID binding to the membrane of the SCs was verified by imaging of the SCs after addition of CTZ (Fig. 5b). Bioluminescence emission localized to the SCs' membrane was evident when imaging without laser excitation, and validated the binding and activity of Gluc (Supplementary Movie 1).

Next, the activation of membrane recruitment of mRFP1-sspB-Nano protein to individual SCs was quantified using fluorescent microscopy. For the purpose of following individual cells before and after substrate addition, we immobilized the SCs on the slide by adding a biotinylated lipid (DOPE-PEG4-biotin) to the SC membrane and coating the microscope slides with streptavidin. SCs remained bound to the surface after multiple additions and mixing of substrate. Activation of iLID and Gluc-iLID was tested using either 488 nm laser or by addition of CTZ. The normalized mRFP1 intensity in SCs with membrane-bound Gluc-iLID increased by 2.5-times after 4 additions of CTZ in comparison to the initial dark conditions (Fig. 5c, green triangles, Supplementary Movie 2). SCs with membrane-bound iLID (without Gluc) were not activated by addition of CTZ (Fig. 5c, black

circles), but demonstrated a 1.5-times increase in mRFP1 signal after 4 min of blue laser activation (Fig. 5c, pink squares).

The difference in the activation levels between iLID and Gluc-iLID SCs can be associated with the addition of Gluc to the N-terminal side, that placed the iLID protein further away from the membrane and increased its exposure to the external solution. It was previously shown that conjugation of iLID to a membrane reduced its sspB-Nano binding ability in comparison to unconjugated iLID due to steric hindrance[49]. Hence, the addition of Gluc, which also acts as a membrane-distancing anchor, improved this issue. This was further validated by activation of Gluc-iLID with blue laser light, which resulted in an even higher increase in mRFP1 intensity, with more than 8-times difference between light and dark conditions, compared to a 1.5-times light-to-dark difference in the original iLID (Fig. S16). Taken together, these results demonstrate efficient and controlled bioluminescence-activated membrane recruitment of sspB-tagged proteins to Gluc-iLID labeled SCs.

## Discussion

Developing an array of signaling and communication pathways is of high importance in SCs to improve their integration with live

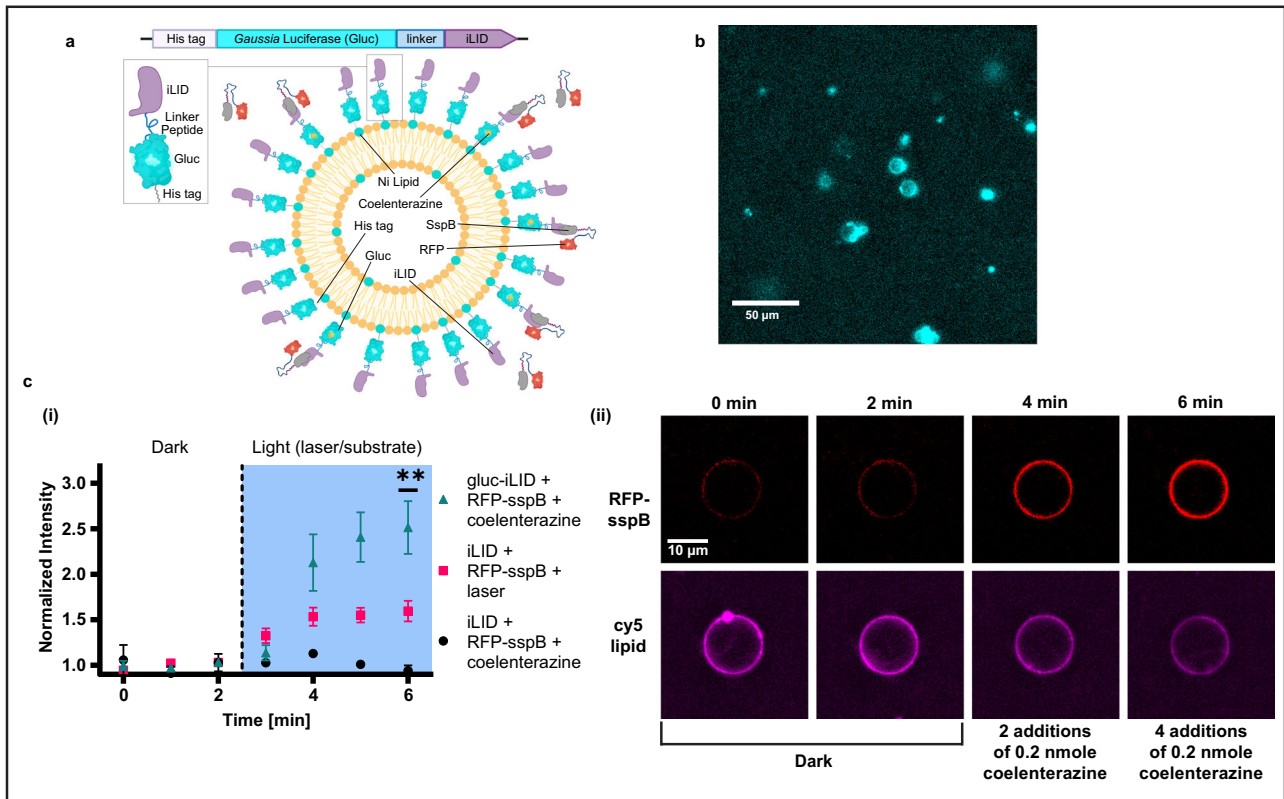

**Fig. 5 Bioluminescence-activated membrane recruitment in synthetic cells. a** A block diagram of the *Gaussia* luciferase (Gluc)-iLID fusion protein elements. Below, a schematic representation of protein recruitment to the synthetic cell membrane by hetero-dimerization of the fusion protein Gluc-iLID with RFP-sspB. **b** Microscopy image of luciferase light emission from synthetic cells with membrane-bound Gluc-EL222 after coelenterazine addition. This experiment was reproduced $n = 5$ times. **c** (i) Membrane recruitment of RFP-sspB with iLID or Gluc-iLID using 488 nm laser illumination or by addition of coelenterazine to activate the bioluminescent reaction. RFP intensity is normalized to the average intensity of each cell in the dark conditions. Data is expressed as a mean ± s.e.m. Welch's two-tailed *t*-test *P*-values; **$p = 0.0092$, ***$p = 0.0001$, for comparison of Gluc-iLID vs. iLID activated by laser and iLID supplemented with coelenterazine at $t = 6$ min, respectively. (ii) Representative single-cell images of RFP-sspB recruitment to a synthetic cell with membrane-bound Gluc-iLID in the dark and after two and four doses of 0.2 nmol coelenterazine (Top row). Bottom row displays the fluorescence of the synthetic cell's DOPE-Cy5 lipid that composed 0.5 mol% of the membrane ($n = 3$ for iLID + coelenterazine, $n = 11$ for iLID + laser, $n = 14$ for Gluc-iLID + coelenterazine). Source data are provided as a Source Data file.

cells and tissues, and build complex control circuits[50,51]. We utilized bioluminescent signaling for intracellular self-activation of SCs and for intercellular communication between synthetic and natural cells. By optimizing the membrane composition and expression of Gluc in SCs, we produced sufficient levels of time-integrated intensity to activate photoconidiation in an adjacent population of fungal *T. atroviride* cells. Moreover, to overcome light intensity limitations, transmitter-responder fusion proteins were designed to control transcription and membrane recruitment in SCs using BRET, by activation of responder proteins with high light intensity demands.

Light is a powerful tool for controlling cellular processes in natural and SCs alike due to the growing availability of genetically encodable optogenetic tools. For in-vivo applications, tissue attenuation of blue light either necessitates using longer wavelengths, for example far red, or intrusive insertion of light sources that can deliver light close to the target site. While engineered red-shifted light-responding proteins or upconversion nanoparticles can be used to avoid using blue-light illumination, having a palette of spectral options is highly advantageous for developing sophisticated and multifunctional SCs[52–54]. Moreover, these methods have some limitations such as restricted spectral shifting and possible changes in activity in the case of red-shifted proteins, tissue overheating and biodistribution challenges when using upconversion nanoparticles.

Here, we have shown that the light generated by Gluc-expressing SCs is sufficient to activate sensitive proteins such as retinal rhodopsins (as the generated light can be seen with the naked eye) and biochemical pathways that are able to integrate photons over time, as in the case of *T. atroviride*. This demonstrated the capability of utilizing bioluminescence for intercellular communication between synthetic and natural cells. Expanding the use of this mechanism to facilitate communication between SCs and other cell types, including eukaryotic cells with natural or engineered light-responsive elements, holds promise as a specific and tunable tool to control cell activities and successfully integrate SCs into a target tissue.

The phospholipid membrane plays an important role in the light interactions of the SC. This was especially evident in the UV-B and UV-C ranges, where we demonstrated the phospholipid membrane's capability to protect DNA from radiation induced-damage. From an evolutionary aspect, these results might hint that lipids possibly had an evolutionary role as UV protecting agents in the prebiotic world which lacked an ozone layer and had high radiation levels[55]. Moreover, the lipid composition was shown to modify the light transmission through the membrane. Therefore, light intensities going in and out of the SC can be tuned by selecting the appropriate membrane components. Interestingly, the lipids' hydrocarbon tail length was previously shown to effect the membrane's permeability to small molecules

in a similar manner to that observed for light penetration in this study[56].

SCs emit diffused light, in contrast to the focused light of lasers and LEDs, that are used traditionally to activate light-sensitive proteins in lab settings. Thus, close proximity between the SC and activated cell is required to compensate for this difference and achieve high local light intensity. Nonetheless, further improvements in the light-generating and light-responding elements are necessary to broaden the range of viable targets[57]. Possible avenues to increase light intensity include improving the enzyme and substrate performance (including luciferases with photon emission in other parts of the spectrum), further optimizing the SC membrane for light transmission, and amplifying or focusing the generated signal. Moreover, engineering of the light-responsive proteins to increase their photo-sensitivity will contribute on the receiving end to enable this synthetic-natural cell cross-talk[58].

Notably, using BRET to facilitate intracellular signaling in SCs enabled a stronger photo-response by efficiently harnessing the energy generated by the bioluminescent reaction. These engineered self-activating proteins are advantageous because they can also be used for simultaneous imaging during the activation process, providing essential information on their activity site. In this study, BRET was used in SCs to facilitate bioluminescent signaling-mediated transcription and membrane recruitment. The controlled transcription mechanism, for instance, could provide a possible tool for achieving precise temporal and spatial resolution of therapeutic protein expression in-vivo. Nevertheless, it is important to note that the current EL222 system exhibits a certain level of leakage and basal expression in the dark, which needs to be addressed if exact expression times and protein levels are required. In addition, while in this system CTZ was supplied externally for initiation of the reaction, further exploration and incorporation of its biosynthesis pathway will enable the next generation of SCs to contain the complete signaling apparatus. This would allow SCs to be used as independent light sources with therapeutic and diagnostic capabilities within the human body.

## Methods

**Lipids.** 1,2-dioleoyl-sn-glycero-3-phosphocholine (DOPC), 1-palmitoyl-2-oleoyl-sn-glycero-3-phosphocholine (POPC), 1,2-dimyristoyl-sn-glycero-3-phosphocholine (DMPC), 1,2-dipalmitoyl-sn-glycero-3-phosphocholine (DPPC), L-α-phosphatidylcholine, hydrogenated (Soy) (HSPC), 1,2-dioleoyl-sn-glycero-3-phosphoethanolamine (DOPE) and 1,2-dioleoyl-sn-glycero-3-phospho-(1'-rac-glycerol) (sodium salt) (DOPG) were purchased from Lipoid (Germany). 1,2-dioleoyl-sn-glycero-3-[(N-(5-amino-1-carboxypentyl) iminodiacetic acid) succinyl] (nickel salt) (DGS-NTA(Ni)) was purchased from Avanti Polar Lipids (USA). Cholesterol was purchased from Sigma-Aldrich (Israel). DOPE-PEG4-biotin and DOPE-cy5 were synthesized by reacting DOPE with NHS-PEG4-biotin or NHS-cy5 (BDL pharma, China).

**Liposome preparation and absorbance measurements.** Liposomes were prepared using the ethanol injection method. Lipids were weighed and dissolved in absolute ethanol and subsequently injected into calcium-free Dulbecco's phosphate buffer saline (PBS; Sigma-Aldrich) preheated to 65 °C reaching a final lipid concentration of 50 mM. The liposomes were extruded five times using a high-pressure Lipex extruder (Northern Lipids, Canada) through 400, 200, and 100 nm polycarbonate-etched membrane (Whatman, Newton, MA, USA) at 65 °C.

For the absorbance measurements, liposomes were diluted 50-fold and measured in a UV-star 96-well microplate (Greiner bio-one, Germany) using the Infinite 200PRO multimode reader (TECAN, Switzerland) controlled with the i-control 1.10 software. Based on the absorbance data, a Lorentzian function was fitted for each of the phospholipids using the curve fitting toolbox in Matlab (version 2021b). The Lorentzian function Eq. (2) is:

$$L = a/((x - b)^{2+c}) + d \quad (2)$$

Where the constants a, b, c, and d denote the Lorentzian amplitude, center, width, and offset respectively, and x denotes the Lorentzian variable - photon energy.

## Synthetic cell preparation

*Preparation of bacterial lysate for synthetic cell solutions.* S30 bacterial lysate was prepared as described previously from BL21(DE3) *E. coli* transformed with the T7

polymerase expressing TargeTron vector pAR1219[59]. For *Gaussia* luciferase expressing synthetic cells, DTT and β-mercaptoethanol were excluded from the S30 solution during the lysate preparation.

*Preparation of lipids in mineral oil.* POPC was lyophilized (FreeZone 2.5; Labconco, USA) overnight. POPC and cholesterol were dissolved separately in chloroform at a concentration of 50 mg ml⁻¹ each. 50 μl of each solution was added to 500 μl mineral oil (Sigma-Aldrich) in a glass vial. The mixture was vortexed and then heated at 80 °C for one hour. The obtained lipid oil was stored at room temperature for up to 2 weeks.

*Emulsion transfer method.* Synthetic cell preparation using the emulsion transfer method was performed as previously described with several modifications[59]. The synthetic cells' inner solution was either prepared according to the components lists in Supplementary Tables 2 and 3 or using the *E. coli* S30 extract system for circular DNA (Promega).

*Shaking method.* Synthetic cell preparation using the shaking method was performed as previously described by Gopfrich, et al.[48] 6 mM of 100 nm liposomes were prepared using the thin film hydration method. A thin lipid film of POPC, DOPG, DGS-NTA(Ni), DOPE-PEG4-biotin and DOPE-Cy5 at molar ratios of 73.5:20:5:1:0.5 was hydrated with PBS containing 10 mM MgCl₂ and extruded as described in the liposomes' preparation section. 100 μl of liposomes diluted to 2 mM were mixed with 200 μl of FC-40 oil (FL-0005-HP; Iolitec, Germany) with 10 mM of PFPE–carboxylic acid (Krytox; Costenoble, Germany) and 0.8% of fluorosurfactant (RAN Biotechnologies, USA) and incubated overnight. The bottom layer was then removed and 100 μl of PBS, followed by 100 μl of 1H,1H,2H,2H-Perfluoro-1-octanol (PFO; Sigma-Aldrich), were added on top of the remaining top layer. After 40 min of incubation, the top layer containing the synthetic cells was extracted.

**Percentage of active synthetic cells.** The percentage of active SCs was measured with imaging flow cytometry using the AMNIS ImageStream®X Mk II (Luminex Corporation, USA, controlled with the AMNIS Inspire software, version 200.1.681.0) and analyzed using the IDEAS software (version 6.2). SCs with or without a plasmid DNA expressing super folder GFP (and composed of an internal solution as described in Supplementary Table 2) were incubated for 1 h at 37 °C and diluted 10-fold in the synthetic cells' outer solution (Supplementary Table 4) before their analysis with the ImageStream®X. SCs events were manually verified using images obtained from the brightfield channel, and their activity level was assessed by measuring their GFP fluorescence intensity using the 488 nm laser and the 505-560 nm emission channel.

**UV exposure induced DNA damage.** 10 μg ml⁻¹ of DNA (full sequence available in Supplementary Data 1) in a modified synthetic cell internal solution containing only sucrose, HEPES KOH pH = 8, magnesium acetate, potassium acetate, ammonium acetate and PEG 6000 (in the same final concentrations reported in Supplementary Table 2) was exposed for 20 min to 254 nm UV light (R-52 Grid lamp; UVP, USA) placed 10 cm above a 96-well plate before or after encapsulation in synthetic cells. DNA from the samples exposed to UV before and after encapsulation was extracted and PCR amplified using Phusion polymerase (Thermofisher, USA). The reaction products were then separated on a 1% agarose gel (uncropped and unprocessed image is available in the source data).

**Luminescence assays.** Luminescence was measured using the Infinite 200PRO multimode reader (TECAN) controlled with the i-control 1.10 software. The CFPS or synthetic cells reactions were mixed in a 1:1 ratio with native coelenterazine (Nanolight, USA) just prior to the measurement.

*Luciferase comparisons.* For comparison of light production in Rluc and Gluc CFPS reactions with the unmodified internal (reducing) conditions, reactions were incubated at 37 °C and 1200 rpm for 1 h. The reactions were measured after addition of a final concentration of 5 μM CTZ in a 384-well white microplate. For comparison of light production in Rluc synthetic cells and Gluc synthetic with the modified internal (oxidizing) conditions, synthetic cells were incubated at 37 °C for 1 h. 40-fold diluted synthetic cells and CTZ in a final concentration of 5 μM were mixed and measured in a 384-well white microplate.

*Comparison to commercial transcription-translation system.* Gluc light emission was compared in CFPS reactions and SCs composed of either the PURExpress system supplemented with disulfide bond enhancer (NEB, USA) prepared according to the manufacturer's protocol or the self-prepared internal solution (described in Supplementary Table 3). For encapsulation in SC, the commercial internal solution was supplemented with 200 mM sucrose and PEG 6000 (3% w/w) to adjust the solution's density and allow generation of vesicles. All reactions were incubated at 37 °C for 1 h. Before the luminescence measurement, CFPS reactions were diluted 1000-fold and synthetic cells solutions were diluted to reach equal absorbance of 0.1 at OD400 to ensure similar cell density. The diluted samples were

measured after addition of a final concentration of 5 µM of CTZ in a 384-well white microplate.

*Luciferase incubation temperature comparison.* Gluc expressing synthetic cells were generated and split into tubes incubated at either 30 °C or 37 °C. Following 1 h of incubation, the samples were measured after addition of a final concentration of 5 µM of CTZ in a 384-well white microplate.

*Luciferase production kinetics.* Gluc expressing synthetic cells were incubated at 37 °C. At each time point, 15 µl of synthetic cells were taken and diluted 4-fold in PBS. The diluted sample was measured after addition of a final concentration of 2.5 µM of CTZ in a 384-well white microplate.

*Luciferase kinetics and re-activation of light emission.* Gluc-expressing synthetic cells were incubated at 37 °C for 1 h. For measuring the luciferase reaction kinetics, the synthetic cells were subsequently diluted 400-fold, mixed with a final concentration of 100 µM CTZ and measured in a 96-well white microplate. Luminescence was measured every 30 s for 15 min. For luminescence re-activation measurements, the synthetic cells were diluted 4-fold and mixed with 0.125 nmol of native CTZ in a 384-well white microplate. After measuring luminescence in one-minute intervals for 5 min, a second dose of 0.125 nmol of native CTZ was added to the same well and luminescence was measured again for 5 min.

**Synthetic cell size and concentration measurements**. Size analysis of synthetic cells was performed by light diffraction using the Mastersizer 3000 (Malvern Instrument, UK). The synthetic cell concentration was measured by imaging flow cytometry using the AMNIS ImageStream®X Mk II (Luminex Corporation, USA) and analyzed using the IDEAS software (version 6.2). Synthetic cells were diluted 10-fold in the synthetic cells' outer solution (Supplementary Table 4) and counted based on the number of events verified as synthetic cells using the images from the brightfield channel. The synthetic cell concentration was calculated by dividing the number of verified synthetic cells by the total volume of sample analyzed.

**Cryogenic scanning electron microscopy imaging**. Cryogenic scanning electron microscopy (cryo-SEM) imaging was performed using Zeiss Ultra Plus high-resolution SEM, equipped with a Schottky field-emission gun and with a BalTec VCT100 cold-stage maintained below −145 °C. Specimens were imaged at low acceleration voltage of 1 kV, and working distances of 3–5 mm. Both the Everhart Thornley ("SE2") and the in-the-column ("InLens") secondary electron imaging detectors were used. The energy-selective backscattered ("ESB") detector was used for elemental contrast between the organic and the aqueous phases. Low-dose imaging was applied to all specimens to minimize radiation damages.

Specimens were prepared by the drop plunging method, a 3 µL drop of solution was set on top of a special planchette maintaining its droplet shape and was manually plunged into liquid ethane, after which it was set on top of a specialized sample table. The frozen droplets were transferred into the BAF060 freeze fracture system, where they were fractured by a rapid stroke from a cooled knife, exposing the inner part of the drop. They were then transferred into the pre-cooled HR-SEM as described above. Ideally, imaging was performed as close as possible to the drop surface, where cooling rate should be maximal.

**Western blot analysis**. Synthetic cell samples after protein production were diluted 8-fold and analyzed using SDS-PAGE with a 12% acrylamide gel. The gel was blotted onto a nitrocellulose membrane and blocked with 5% nonfat milk powder in Tris-buffered saline. Gaussia luciferase Polyclonal Antibody (PA1-181, Invitrogen, USA) diluted 1:3750 in Tris-buffered saline with 0.5% Tween-20 and 0.5% nonfat milk powder was incubated with the membrane overnight at 4 °C. After washing, the blots were incubated with horseradish peroxidase-conjugated goat anti-rabbit secondary antibody (ab6721, Abcam, UK) diluted to 1:20,000 and developed with Clarity Western ECL Blotting Substrate (BioRad, USA). The results were imaged using the Fusion FX6 imaging system (Vilber, France) controlled with the Evolution-capt software (version 18.02). Uncropped and unprocessed image is available in the source data.

For quantification of Gluc production, analysis of the images was performed with ImageJ gel analysis plug-in and use of a calibration curve for Gluc in known concentrations.

**Photo-activation of conidiation in fungi with synthetic cells**
*Fungal growth.* A *Trichoderma atroviride* inoculum was plated on the center of a PDYC (24 g l⁻¹ potato dextrose broth (Difco, UK), 1.2 g l⁻¹ casein hydrolysate (Sigma-Aldrich), 2 g l⁻¹ yeast extract (Difco) agar plate and incubated for twenty-four hours in dark conditions. 3 ml of PDYC media were subsequently added to a new 10 cm culture plate. In the center of the plate a Whatman 50 filter paper cut to a diameter of 9 cm was placed over an 8 cm Whatman 1 filter paper. On top of these, a 0.5 cm square from the periphery of the fungal growth on the agar plate was placed in the center and incubated in the dark at room temperature for 48 h.

*Exposure to synthetic cells.* Synthetic cells were produced using the emulsion transfer method and diluted to the desired cell concentration. The synthetic cell concentration was quantified by flow cytometry using the AMNIS ImageStream®X Mk II as described above. Two consecutive exposures of the fungi to synthetic cells were performed. In each exposure, 500 µl of synthetic cells were mixed with 500 µl of native CTZ in a final concentration of 100 µM in a transparent plastic bag. The bag was localized over the periphery of the fungal colony in one part of the plate for 15 min and then removed. The plates were left for incubation in the dark for an additional period 24 h, after which they were imaged using a regular camera.

*Image analysis.* Background normalization of the images was performed manually in ImageJ (version 2.0.0) to achieve the same background average pixel value for all images[60]. The images were then converted to grayscale and then to black and white using a pixel threshold value equal to 45. The percentage of black pixels in a rectangle containing the top part of the plate where the synthetic cells were placed was calculated separately for each image, using python (version 3.7).

*Generation of a light dose response curve.* A dose response curve of *T. atroviride* sporulation was generated using a blue LED lamp (F-PLS10W/B 10 W, Eurolux). After growing the fungal colonies for 48 h in the dark, each colony was exposed to a specifically measured light dose. The total light dose ranged from 10 to 22500 µE m⁻² as measured using a quantum radiometer photometer (Model LI-185B with a Quantum sensor, LI-COR, USA) by changing the exposure time of the colony (from 2 s to 15 min). After an incubation period of 24 h, the fungal colony-containing filter papers were transferred briefly to 70% ethanol and then fixed in absolute ethanol. Spores were isolated and suspended in 5 ml of water and their optical density was measured at 600 nm. One OD 600 unit corresponds to approximately 10⁷ spores. A Michaelis-Menten curve was fitted using the least square regression method in Prism GraphPad. For analysis of the SC treatment, fungal colonies were treated as described in the "Exposure to synthetic cells" section above, followed by fixation and spore isolation to measure scatter.

**Transcription activation in synthetic cells**
*LED illumination system.* Five 470 nm blue LEDs (C503B-BAN-CY0C0461; Mouser, Israel) were connected together in series using an Arduino microcontroller and an external power supplier 0–30 V. On-off intervals were set by controlling relay modules using the Arduino IDE software. Blue light intensity was measured with a S310C light sensor (Thorlabs, USA).

*EL222 mediated activation of transcription in cell free reactions.* Cell-free reactions based on the internal synthetic cell solution (Supplementary Table 2) for Rluc expression or *E. coli* S30 extract system for circular DNA (Promega) for RFP expression were supplemented with 2.5 µM EL222 (unless stated otherwise) and incubated at 37 °C in 384-well microplates coated with an adhesive film to prevent evaporation. LEDs were placed 7 cm above the plate, providing approximately 12 W m⁻², with on-off intervals of 20 and 70 s. For analysis of Rluc expression, native CTZ was added to a final concentration of 5 µM and luminescence was measured. For analysis of RFP expression, fluorescence intensity was measured with excitation / emission at 540 nm / 600 nm.

*EL222 activation in synthetic cells.* Synthetic cells with an internal solution based on the *E. coli* S30 extract system for circular DNA (Promega) were supplemented with 2.5 µM EL222. Prior to their incubation, each synthetic cell batch was divided into dark and light groups, both incubated at 37 °C in 384-well microplates coated with an adhesive film under dark or blue-light conditions. LEDs were placed 2.8 cm above the plate, providing approximately 19 W m⁻², with on-off intervals of 20 and 70 s.

Self-activating synthetic cells based on the *E. coli* S30 extract system for circular DNA (Promega) were supplemented with 2.5 µM of Gluc-EL222. Prior to their incubation, each synthetic cell batch was divided into dark and light groups, both incubated at 37 °C. Native CTZ (0.2 nmol) was added every 30 min to the light group for a total of four times. The samples were further incubated for 90 more minutes after the last substrate addition. Just prior to the final fluorescence measurements, CTZ was added to the dark group in the same concentration that was added to the light group in order to eliminate differences due to substrate auto-fluorescence.

**Membrane recruitment in synthetic cells**
*Membrane recruitment of RFP-sspB-Nano.* A PDMS-walled chamber (Sylgard 184; Dow, USA), 5.5 mm in diameter and 2.5 mm height, was placed on a 22 × 50 mm deckglaser glass slide (Marienfeld, Germany). The bottom of the chamber was coated with 10 µg ml⁻¹ streptavidin (Promega) overnight at 4 °C, or with 1% bovine serum albumin (Sigma-Aldrich) for 1 h at room temperature. The chamber was washed with PBS and in case of streptavidin coating, was coated once more for 1 h with 5 mg ml⁻¹ of cold water fish skin gelatin (Sigma-Aldrich) and washed again with PBS.

Synthetic cells were prepared using the shaking method (see synthetic-cell preparation), and 100 nM of his-iLID or his-Gluc-iLID were added to the solution and incubated for 1 h at room temperature and shaking of 100 rpm. Under dark

conditions, 25 nM of RFP-sspB-Nano were added to the cells and placed in the coated chamber in dark conditions for 30 min.

Imaging was performed using a standard inverted microscope (Eclipse Ti2; Nikon, controlled with NIS-Elements AR, version 5.11.01) outfitted with a 60×1.4 NA objective lens (Nikon). Low power laser illumination of approximately 5 mW cm$^{-2}$ 640 nm and 500 mW cm$^{-2}$ 561 nm lasers for imaging of Cy5 and mRFP1 respectively were used. Images were captured on a Sona sCMOS detector (Andor, Northern Ireland). Blue light laser illumination was performed with 488 nm laser at approximately 20 mW cm$^{-2}$ with on-off intervals of 1.25 and 3.75 s for one minute. For recruitment in synthetic cells functionalized with his-Gluc-iLID, native coelenterazine (0.2 nmol) was added every minute for a total of four minutes.

*Gluc-iLID light emission imaging.* Synthetic cells functionalized with Gluc-iLID were imaged after addition of 100 µM substrate (final concentration) using a standard inverted microscope (Eclipse Ti2; Nikon) outfitted with a 40×0.75 NA objective lens (Nikon) and equipped with iXON EMCCD camera (Andor). Images were obtained with 400 msec exposure time and gain 300.

**Statistics**. The statistical analysis including student's *t*-test, one-way, two-way ANOVA and Michaelis–Menten model fitting was performed using Prism GraphPad version 8.3.0.

**Reporting summary**. Further information on research design is available in the Nature Research Reporting Summary linked to this article.

## Data availability
All data supporting the findings and conclusions of this study are available within the paper, its Supplementary Information files and source data file. All other relevant data are available from the corresponding author upon reasonable request. Source data are provided with this paper.

## Code availability
The codes used in this study for image analysis and fitting of the spectral line shape Lorentzian have been deposited in Zenodo under the following https://doi.org/10.5281/zenodo.6368147[61].

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

## Acknowledgements

This project received funding from the European Union's Horizon 2020 research and innovation program under the grant agreement No 680242-ERC-[Next-Generation Personalized Diagnostic Nanotechnologies for Predicting Response to Cancer Medicine]. The authors also acknowledge the support of Israel Innovation Authority for the Nofar Grant (67967), the Israel Science Foundation (1778/13, 1421/17); The Israel Ministry of Economy for a Kamin Grant (52752, 69230); the Israel Ministry of Science, Technology & Space (3-16963; 3-17418); the Ministry of Agriculture & Rural Development - Office of the Chief Scientist (323/19); the Israel Cancer Association (2015-0116); the Leventhal 2020 COVID19 Research Fund (ATS #11947), the German-Israeli Foundation for Scientific Research and Development for a GIF Young grant (I-2328-1139.10/2012); the European Union FP-7 IRG Program for a Career Integration Grant (908049); the Phospholipid Research Center Grant (ASC-2018-062/1-1); the European Research Council (ERC) under the European Union Horizon 2020 research and innovation program to Y.S. (802567); the European Research Council (ERC) to L.G. (ERC-2017-COG-773181-iPS-ChOp-AF); the Louis family Cancer Research Fund, a Mallat Family Foundation Grant; the Unger Family Fund; a Carrie Rosenblatt Cancer Research Fund, the Technion Integrated Cancer Center (TICC), the Russell Berrie Nanotechnology Institute, and the Lorry I. Lokey Interdisciplinary Center for Life Sciences & Engineering. A.S. acknowledges the Alon and Taub Fellowships. O.A. acknowledges the Jacobs and Gutwirth Fellowships. L.E.W. and Y.S. acknowledges the Zuckerman foundation, G.C. acknowledges the "Baroness Ariane de Rothschild Women Doctoral Program" from the Rothschild Caesarea Foundation. The Max Planck Society is appreciated for its general support. Mia R. Albalak and Ravit Abel contributed equally to this manuscript. The authors thank Mrs. Naama Koifman from the Technion Center for Electron Microscopy of Soft Matter (TCEMSM) for performing the cryo-SEM imaging, and Mr. Nadav Opatovski for his technical assistance with microscope calibration. The authors thank Dr. Assaf Zinger (Department of Chemical Engineering, Technion), Dr. Shai Berlin, Mr. Michael Andreyanov (Department of Medicine, Technion), Prof. Ygal Rotenstreich, Dr. Ifat Sher, Mrs. Zehavit Goldberg (Sheba Medical Center, Israel) and Mr. Yahli Holtzman for their professional input and helpful discussions that greatly contributed to the paper. Illustrations in this paper were created with BioRender.com and Adobe Illustrator.

## Author contributions

O.A. and A.S. conceived of the approach. O.A., M.R.A., R.A., L.E.W., A.G., O.S., and B.A.H. designed the experiments, O.A., M.R.A., R.A., L.E.W., G.C., A.G., and B.A.H. performed the experiments. Y.K. and I.K. developed the spectral line shape fitting model. All authors contributed to data analysis, provided substantial input to all aspects of the project and participated in the preparation of the manuscript.

## Competing interests

O.A. and A.S. are listed as inventors on a provisional patent application disclosing materials and formulations related to the presented results filed with the US Patent and Trademark Office (patent application number: US 63/184,876, patent applicant: Technion Research & Development Foundation Limited). The remaining authors declare no competing interests.
