## [Peer Review File · Nature Communications]

Reviewers' Comments:

Reviewer #1:

Remarks to the Author:

The manuscript by Adir and colleagues describes blue light controlled synthetic minimal cells.

The article is well written, the rationale is clear and conclusions are convincing and supported by the available data.

It would be valuable to have better characterization of the yields and efficiency of the system. What percent of synthetic cells is "active" in the experiments? What is the efficiency of liposome loading?

Finally, what percentage of fungi cells respond to the stimuli?

On figure 4b, the difference between no DNA control and luciferase sample without EL222 indicates that there is some break through expression. This should be addressed in the discussion. Since it's a negative feature of the system, not disqualifying the data but it should be addressed.

Why were the experiments run at 37C, when S30 extract is most active at slightly lower temperatures?

Why was *Trichoderma atroviride* used as model live cell to activate?

Minor comments:

it would be useful to explain the rationale behind selecting 480nm light, instead of lower energy, more penetrable and less harmful to DNA green or red shifted light.

Was it only because of availability of proteins, or was there another experimental reason?

I would hesitate to agree that Luciferase induced activation of fungi is similar to quorum sensing response. Quorum sensing is chemical signal based. While what Authors observe is a very impressive phenomena, it's more like a dose response than quorum sensing.

Similarly, I'm not sure if term "optogenetic" can be used here. Not every biological process that's light dependent automatically becomes optogenetics.

Reviewer #3:

Remarks to the Author:

In this work, the authors demonstrate the development of light-emitting "synthetic cells", which are lipid vesicles containing a transcription-translation mix for the synthesis of a luciferase. Upon substrate addition, bioluminescence is observed. They apply the light emitted from the vesicles to induce conidiation in fungi, a process known to be induced already at low levels of light. They further construct fusion proteins between luciferase and a light-responsive transcription factor (EL222) or a light-inducible dimerizer. They use this to control gene expression and membrane-recruitment in response to the addition of luciferase substrate, respectively.

The use of luciferases to activate optogenetic tools has been established previously in different configurations. For example, [10.3389/fnbeh.2014.00108](https://doi.org/10.3389/fnbeh.2014.00108) and [doi: 10.3389/fnbeh.2014.00108](https://doi.org/10.3389/fnbeh.2014.00108) used luciferase to control neuronal activation engineered for light-sensitive ion channels.

[doi:10.3390/life10120318](https://doi.org/10.3390/life10120318) is a review article on using luciferases to induce optogenetics for example of light-sensitive dimerizers to induce recruitment of proteins. The following works describe similar systems among which a fusion of luciferase to a light-regulated transcription factor: <https://doi.org/10.1364/BRAIN.2017.BrTu2B.1> and

<https://doi.org/10.1364/BRAIN.2020.BW4C.1>, DOI: [10.7554/elife.43826](https://doi.org/10.7554/elife.43826) and DOI: [10.1038/s41467-021-20913-1](https://doi.org/10.1038/s41467-021-20913-1).

Specific comments:

The conceptual advance of the manuscript with regard to the previous work (showing cell-to-cell

communication, transcriptional modulation and recruitment via luciferase-photoreceptor pairs) is not clearly described. The main story seems to be that the luciferase is produced within a lipid vesicle.

Measuring the absorbance of light by membranes does not contribute too much new information. From the structure of the lipids it is expected/known that significant absorbance occurs at short wavelengths.

Similarly, the paragraph on protection of DNA from UV light by the membranes seems off-topic and deviates the reader from the main message of the manuscript. The detailed absorbance measurements and DNA protection should rather be separated from the rest of the manuscript.

Point-by-point Response - NCOMMS-21-23423-T

Synthetic Cells with Self-Activating Optogenetic Proteins Communicate with Natural Cells

Summary

We would like to thank the editor and referees for their thorough review and constructive remarks that improved the study in its conceptual and experimental aspects. We revised the manuscript accordingly, adding synthetic cell characterization, synthetic-natural cell communication experiments, and mathematical functions for the lipid-light interactions to address these questions. Below, are the reviewer comments (in bold) and our point-to-point responses to each of the comments. Cited parts from the manuscript are in blue font and their location in the main text is detailed. Changes made to the manuscript files are highlighted in **green**.

Reviewer 1:

- 1. It would be valuable to have better characterization of the yields and efficiency of the system. What percent of synthetic cells is "active" in the experiments? What is the efficiency of liposome loading?**

We thank the reviewer for this important comment. To address this, we produced GFP in our synthetic cells (SCs) and used imaging flow cytometry to identify active vs. inactive cells based on their fluorescence intensity (using SCs without DNA as a negative control for background auto fluorescence). The results are presented in Fig. 1e and detailed in Page 5, line 158: "*Imaging flow cytometry was then used to assess the percentage of active SCs (executing the transcription and translation processes), by measuring the fluorescence intensity of SCs encapsulating a CFPS solution and GFP encoding DNA (Fig. 1e). 85% of the SCs were active and displayed higher fluorescence levels than the maximal background auto-fluorescent signal that was measured using a population of SCs without a DNA template.*"

The SCs' content is composed of more than 30 synthetic materials as well as biologically derived bacterial lysate, each having different physical and chemical properties and thus different loading efficiencies. Therefore, measuring the percentage of active cells also represents the overall efficiency of loading to drive the transcription and translation.

Figure 1e. The percentage of active synthetic cells was measured using imaging flow cytometry of GFP-expressing synthetic cells (analyzed n=331 synthetic cells without DNA and n=308 synthetic cells with GFP DNA). Frequency was normalized to the total number of cells in each sample.

2. Finally, what percentage of fungi cells respond to the stimuli?

We thank the reviewer for this important question. To address this, we measured the light response of the fungal colonies to a range of light intensities generating a light-response curve and asked what fluence (photons per unit area) of light reaching the fungal colony from a distance would correspond to the light emitted by the synthetic cells. We then calculated the activation level generated by the SCs treatment in comparison to the maximum response upon reaching saturation. The results are presented in Fig. 3e and detailed in Page 9, line 292:

"In order to estimate the conidiation activation levels of the fungal colonies more precisely, a light dose calibration curve ranging from $10 \mu\text{E m}^{-2}$ to $22500 \mu\text{E m}^{-2}$ was generated using a blue LED lamp (Fig. 3e, black circles). The sporulation level in each treatment was measured by suspending the spores from the colonies in water and measuring the scatter at 600 nm. A Michaelis-Menten model was used to fit the measured data and to calculate the equivalent light dose and activation level of the SC treatment. The Michaelis Menten equation (1):

$$(1) OD_{600} = (V_{max} \cdot \text{Total Light Dose}) / (K_m + \text{Total Light Dose})$$

provided best fit values of 0.4157 (absorbance units at 600 nm) for V_{max} (denoting the optical density at saturation) and $2407 \mu\text{E m}^{-2}$ for K_m (denoting the light dose required to reach 50% of the saturation level) with $R^2=0.9582$. The calculated light dose for SCs was approximately $4552 \mu\text{E m}^{-2}$ (fig. 3e, pink square), which is higher but in the 95% confidence interval (CI) range of the K_m of the curve (95% CI of 1186 to $5253 \mu\text{E m}^{-2}$). Using the best fit V_{max} value of the curve, the activation level of the SC treatment was estimated to be approximately 65% of the saturation level."

Figure 3e. A light dose-response calibration curve was generated in *Trichoderma atroviride* using a blue LED (black circles). A Michaelis-Menten curve was fitted to the data (dotted pink line, $R^2=0.9582$) and the total light dose of the synthetic cell treatment was calculated according to the fitted model (pink square). Data is expressed as mean \pm standard deviation (between $n=2$ to $n=3$ independent samples).

Thanks to the reviewers' suggestion we now quantified a very important aspect of the system and are therefore able to provide a more detailed description of the SCs' influence on fungal cells, highlighting the strength of our approach.

3. **On figure 4b, the difference between no DNA control and luciferase sample without EL222 indicates that there is some break through expression. This should be addressed in the discussion. Since it's a negative feature of the system, not disqualifying the data but it should be addressed.**

We thank the reviewer for this comment. The presence of basal leakage expression in the EL222 system is now clarified in the discussion, Page 15 line 473: "The controlled transcription mechanism, for instance, could provide a possible tool for achieving precise temporal and spatial resolution of therapeutic protein expression *in-vivo*. Nevertheless, it is important to note that the current EL222 system exhibits a certain level of leakage and basal expression in the dark, which needs to be addressed if exact expression times and protein levels are required."

4. **Why were the experiments run at 37C, when S30 extract is most active at slightly lower temperatures?**

Thank you for this question. We now added an analysis of the expression level of *Gussia luciferase* in 30°C and 37°C in our system and the results are now presented in Fig. S8. Importantly, we look

towards implementing this system in biomedical settings in the future, and testing its functionality at physiological temperature imitates these conditions more closely.

Supplementary Figure 8. The effect of incubation temperatures on Gluc expression in SCs. Comparison of light emission from Gluc expressing SCs after incubation at 30°C or 37°C. Data is represented as the mean ± standard deviation (n=3 independent samples). Nested two-tailed t-test P-value; **P=0.0034.

5. Why was *Trichoderma atroviride* used as model live cell to activate?

Trichoderma was selected as the live cell model in the SC-to-natural cell signaling experiments because it responds to blue light which fits the wavelength generated by the Gaussia luciferase expressed in our SCs, and because the high sensitivity of the photo-induced process and its abidance to the Bunsen-Roscoe law of reciprocity enable it to generate more significant response when exposed to the light levels generated by our bioluminescent system.¹ Furthermore, the fungi's endogenous photoreceptor is a light, oxygen, voltage (LOV) domain protein, which is similar in its spectral properties and sensitivity to engineered LOV domain photoreceptors used in other optogenetic applications. In terms of the measured outcome, sporulation is readily recognized and measured using both image analysis and optical density measurements. This is now clarified in the text, page 9, line 269:

"To demonstrate intercellular signaling between light-producing SCs and light-responsive natural cells we employed the soil fungus *Trichoderma atroviride*. Two blue-light regulator (BLR-1, a LOV-domain protein, and BLR-2) proteins control the photo-activation of this fungus in response to blue light, triggering several downstream processes including conidiation and the expression of the DNA repairing photolyase enzyme^{1, 2, 3}. The level of fungal conidiation depends on the total light exposure with no dependency on the light intensity or duration (abides the Bunsen-Roscoe law of reciprocity), and thus fungal cells can be activated with continuous low-intensity light emission.¹"

- 6. it would be useful to explain the rationale behind selecting 480nm light, instead of lower energy, more penetrable and less harmful to DNA green or red shifted light. Was it only because of availability of proteins, or was there another experimental reason?**

Blue light was selected for several reasons. In terms of luciferase light emission, currently the brightest luciferases known emit light in the blue range (Gaussia luciferase, NanoLuc), and meet our requirement for a system a very high light yield in order to activate a photo-responsive protein. In addition, there are several families of light-responsive proteins that respond to blue light, including the light, oxygen, voltage (LOV) domain family and the cryptochrome families that bind Flavin mononucleotide (FMN) and falvin adenine dinucleotide (FAD) as their chromophores respectively. These protein families provide multiple possible targets for activation.⁴ Another consideration was that blue light has poor tissue penetration in comparison to red shifted wavelengths. Therefore, developing a SC system that will deliver light into deep tissue is of specific interest, while red-shifted wavelengths can still be delivered from an exterior source.

We now address this issue on page 4, line 120:

"In this work, we focused on interactions with blue light (480 nm), which can be used to activate photoreceptors such as light-oxygen-voltage-sensing (LOV) domains present in many photoactivatable proteins, and is emitted by certain types of luciferases, including *Renilla* and *Gaussia* luciferase^{5, 6}. Furthermore, the use of blue light for *in-vivo* applications traditionally poses a challenge for non-implanted sources due to its poor tissue penetration."

and page 14, line 461:

"Possible avenues to increase light intensity include improving the enzyme and substrate performance (including luciferases with photon emission in other parts of the spectrum), further optimizing the SC membrane for light transmission, and amplifying or focusing the generated signal. Moreover, engineering of the light-responsive proteins to increase their photo-sensitivity will contribute on the receiving end to enable this synthetic-natural cell cross-talk".

- 7. I would hesitate to agree that Luciferase induced activation of fungi is similar to quorum sensing response. Quorum sensing is chemical signal based. While what Authors observe is a very impressive phenomenon, it's more like a dose response than quorum sensing. Similarly, I'm not sure if term "optogenetic" can be used here. Not every biological process that's light dependent automatically becomes optogenetics.**

Thank you for highlighting this issue. We indeed did not intend to claim that the phenomenon we demonstrated in these experiments meets the biochemical definition of quorum sensing. As you wrote, quorum sensing is based on the cellular ability to sense the accumulation of a signaling molecule in the environment due to increased cell density and contains a positive feedback to initiate

a coordinated response on the population level.^{7, 8} Our intention was to emphasize that in this specific process of communication between SCs and fungi cells, we recognize an important similarity between this process and quorum sensing – the dependency on cell density to activate the process. The use of bottom-up assembly for generating SCs allows constructing cellular processes that do not exist in nature and possibly stretch the boundaries of these definitions; in this case, the use of light instead of a chemical entity for generating a cell-density-based response. The flexibility and modularity of our engineered system is ideally suited for further increasing the complexity of this communicative process by integrative addition of functional modules that might allow full recapitulation of quorum-sensing behaviors between natural and synthetic cells. To clarify this important aspect in the manuscript we corrected the text on page 10, line 308:

"The dependency of *T. atroviride*'s photo-activation on the SC concentration resembles the cell density dependency that is a key characteristic of quorum sensing mechanisms in different species.^{7, 8} Nevertheless, natural quorum sensing mechanisms differ from the signaling mechanism demonstrated here as they are based on chemical entities (and not light) and contain a positive feedback loop to initiate a population coordinated response. Moreover, the use of light as a signaling module in this mechanism has the unique property of enabling signal transmission even when the cell populations are separated by a physical transparent barrier."

Regarding the use of the term 'optogenetics', we agree with the reviewer that not every light-dependent process should be considered as optogenetic by nature. However, to our understanding, optogenetic tools can be defined as genetically encoded tools that respond to light and modulate a cellular state or activity.⁹ The elements that were used for the light-responsive synthetic cells are genetically encoded proteins and were previously recognized as optogenetic tools in several publications (EL222^{10, 11}, iLID^{12, 13}), and therefore we believe that the term 'optogenetic' can be used to describe the system designed in this manuscript as well.

Reviewer 3:

- 1. The use of luciferases to activate optogenetic tools has been established previously in different configurations. For example, 10.3389/fnbeh.2014.00108 and doi: 10.3389/fnbeh.2014.00108 used luciferase to control neuronal activation engineered for light-sensitive ion channels. doi:10.3390/life10120318 is a review article on using luciferases to induce optogenetics for example of light-sensitive dimerizers to induce recruitment of proteins. The following works describe similar systems among which a fusion of luciferase to a light-regulated transcription factor: <https://doi.org/10.1364/BRAIN.2017.BrTu2B.1> and <https://doi.org/10.1364/BRAIN.2020.BW4C.1>, DOI: 10.7554/elife.43826 and DOI: 10.1038/s41467-021-20913-1.**

We thank the reviewer for this comment. We expanded our introduction accordingly, adding the following section on page 3, line 82:

"The use of luciferase-generated bioluminescence for activation of optogenetic proteins, genetically encoded light responsive elements that can modulate cellular states, has been explored in diverse cellular processes, including ion channel activation for neuromodulation and transcriptional control in eukaryotic cells.^{14, 15} Due to the weak intensity of luciferase in comparison to external light sources (lasers and LEDs), many of these applications focused on intracellular cascades, in which the luciferase and the target photo-responsive protein are in close proximity. To maximize the efficiency of activation, fusion proteins of luciferase and the target light responsive protein were designed, harnessing bioluminescence resonance energy transfer (BRET).^{16, 17} Recently, bioluminescent trans-synaptic activation between physically connected neurons was enabled by accumulation of luciferase proteins secreted by pre-synaptic cells into the synaptic space, yielding a local concentration that was sufficient to activate a light sensitive protein in the post-synaptic cells.^{18"}

- 2. The conceptual advance of the manuscript with regard to the previous work (showing cell-to-cell communication, transcriptional modulation and recruitment via luciferase-photoreceptor pairs) is not clearly described. The main story seems to be that he luciferase is produced within a lipid vesicle.**

Thank you for this important comment. We modified the abstract, introduction and discussion to clarify the conceptual and experimental advances of our study (all changes in the manuscript are highlighted **in green**).

In the abstract, page 2, line 34:

"Here, we describe the design and implementation of bioluminescent intercellular and intracellular signaling mechanisms in synthetic cells, dismissing the need for an external light source. First, we

engineer light generating SCs with an optimized lipid membrane and internal composition, to maximize luciferase expression levels and enable high-intensity emission. Next, we show these cells' capacity to trigger bioprocesses in natural cells by initiating asexual sporulation of dark-grown mycelial cells of the fungus *Trichoderma atroviride*. Finally, we demonstrate regulated transcription and membrane recruitment in synthetic cells using bioluminescent intracellular signaling with self-activating fusion proteins."

In the introduction, page 3, line 95:

"Here, we harness bioluminescence to engineer intercellular and intracellular signaling mechanisms in SCs for the purpose of activating cellular processes in both natural and SCs. The SCs were composed of giant unilamellar vesicles (GUVs) encapsulating a bacterial-based, cell-free protein synthesis (CFPS) system (Fig. 1a). To enable intercellular signaling between a SC and a natural cell, light-generating SCs were designed to express high levels *Gaussia* luciferase (Gluc) in order to enhance their light emission intensity. These cells were then used to activate photoconidiation in fungal cells, demonstrating their capacity to control a biological process in a subsequent population of natural cells. Next, intercellular bioluminescent signaling processes were engineered in SCs. In order to utilize light-responsive proteins that required higher intensities, self-activating fusion proteins were engineered by coupling Gluc with photo-responsive proteins, facilitating their activation by BRET. This approach was used to control transcription in SCs using a bioluminescent fusion protein of Gluc and the light-activated transcription factor EL222. Light-controlled activation of membrane recruitment in SCs was achieved as well, with an additional BRET-based signaling mechanism utilizing a fusion protein of Gluc and iLID, that dimerized with a sspB-tagged protein when the bioluminescent reaction was initiated. Altogether, these new SC signaling functionalities present opportunities for bioluminescent activation and control of synthetic and natural cells alike."

In the discussion, page 13, line 417:

"We utilized bioluminescent signaling for intracellular self-activation of SCs and for intercellular communication between synthetic and natural cells. By optimizing the membrane composition and expression of Gluc in SCs, we produced sufficient levels of time-integrated intensity to activate photoconidiation in an adjacent population of fungal *T. atroviride* cells. Moreover, to overcome light intensity limitations, transmitter-responder fusion proteins were designed to control transcription and membrane recruitment in SCs using BRET, by activation of responder proteins with high light intensity demands."

As mentioned above, an important milestone in this study was to engineer optimized light-producing SCs by facilitating post translational disulfide bond formation to achieve correct protein folding and high expression levels of active *Gaussia* luciferase. Besides being a main checkpoint on the way to

achieve the desired intercellular signaling, expression of disulfide bond-containing proteins has value on its own and can be utilized for other proteins with therapeutic or diagnostic properties. To assess our system further, we compared the production of Gaussia luciferase (by measuring light emission) between a commercial and our optimized internal solution. These results are now presented in Fig. S7 and detailed on page 7, line 229:

"Furthermore, these engineered SCs generated a 10-fold great photon emission compared to SCs expressing Gluc using a commercial CFPS reaction (Fig. S7). In contrast, the commercial CFPS solution without encapsulation in SCs outperformed our self-prepared CFPS solution. This difference highlights the importance of optimizing the SCs' internal solution properties (such as density and osmolality) to fit the encapsulation process."

Supplementary Figure 7. Comparison of light emission levels in Gluc expressing cell-free protein expression (CFPS) reactions and synthetic cells between self-prepared and commercial internal solutions. Prior to the luminescence measurements, CFPS reactions were diluted 1000-fold and synthetic cells were all diluted to reach an absorbance of 0.1 at OD400 to ensure similar cell density. Data is represented as the mean \pm standard deviation (n=2 independent samples for the commercial CFPS reactions and n=3 independent samples for all other experimental groups). Nested two-way ANOVA P-value; ****P<0.0001.

3. Measuring the absorbance of light by membranes does not contribute too much new information. From the structure of the lipids it is expected/known that significant absorbance occurs at short wavelengths.

Thank you for this comment. Our measurements provide new information on the differences in light absorbance in different types of phospholipids. We show that the light absorption is correlated with the phospholipid's tail length and saturation level. We didn't find previous mentions of this

phenomenon which has interesting implications since light is used as an activating modality for many applications in lipid vesicles, and therefore the choice of lipids could affect the efficacy of these systems. This includes signaling mechanisms in SCs, light-controlled drug release systems and liposomal systems delivering photosensitizers^{19, 20}.

To understand this finding in more depth and investigate whether this correlation is true for other wavelengths, we scanned the absorbance of different phospholipids in a wider section of the spectrum (230-800 nm). Using this data, we now added a spectral line shape Lorentzian to each of the phospholipids, which enabled us to extract their exact absorption resonance. The results are presented in Fig. S4 and detailed in page 5, line 165:

"Next, we scanned the absorbance spectrum of the tested phospholipids at different wavelengths (230-800 nm) to investigate whether the absorbance correlations recognized for 480 nm light were maintained for other wavelengths that could also be used for light signaling (Fig. S4). The obtained absorbance data of each phospholipid was fitted with a spectral line shape Lorentzian function ($R^2 \geq 0.9993$) to extrapolate its absorbance properties in shorter wavelengths (down to 120 nm). From the measured data and Lorentzian function, we determined that the correlation between the phospholipids' tail length and absorbance continued in other parts of the spectrum. Interestingly, unsaturated phospholipids that had lower absorbance at 480 nm than phospholipids with saturated tails of the same length, demonstrated higher peak absorbance in the range of 160-200 nm according to their Lorentzian function (characteristic of carbon double bonds²¹). Furthermore, all of the liposome formulations exhibited a similar absorbance trend with absorption resonance wavelengths in the UVC range (200-290 nm), and little light attenuation at wavelengths longer than 480 nm (Fig. S4). The high agreement between the measured data and fitted Lorentzian functions indicates that light scattering in these measurements was negligible."

Supplementary Figure 4. The absorbance spectrum of 100 nm liposomes composed of different phospholipids. The measured absorbance spectrum of DOPC, POPC, DMPC, DPPC and HSPC liposomes between 230 and 800 nm is represented in units of wavelength (a) and photon energy (b). Data is represented as a mean (n=5 independent samples for DOPC, n=6 independent samples for POPC, n=3 independent samples for DMPC, DPPC and HSPC). Representation of the absorption in units of photon energy enabled fitting the absorption data of each phospholipid to a Lorentzian function to determine the absorption resonance of each phospholipid. (dotted line, $R^2=0.9995$ for DOPC, $R^2=0.9993$ for POPC, $R^2=0.9993$ for DMPC, $R^2=0.9995$ for DPPC, $R^2=0.9996$ for HSPC).

- 4. Similarly, the paragraph on protection of DNA from UV light by the membranes seems off-topic and deviates the reader from the main message of the manuscript. The detailed absorbance measurements and DNA protection should rather be separated from the rest of the manuscript.**

Thank you for raising this subject and allowing us to clarify this point. Because the manuscript is focused on developing bioluminescence mediated intercellular and intracellular signaling mechanisms for synthetic cells, we investigated light-lipid interactions in the process (elaborated in our response to the previous comment). DNA protection by lipid membranes from UV radiation provides an indication on a possible role of lipid compounds in the prebiotic world, which will be relevant to the readers of the manuscript that are also interested in synthetic cells as model systems for research on the origin of life. Furthermore, UV radiation was used previously to activate processes in synthetic cells and therefore light-membrane interactions and effects on DNA integrity are of interest.²² Therefore, we adjusted the text to clarify this part on page 6, line 181:

"Considering the wide application range of SCs as model systems for various cellular processes, and as diagnostic and therapeutic tools, we further investigated the interaction of the SCs' phospholipid membrane with light, and specifically its implications on the SCs' DNA integrity."

References:

1. Horwitz BA, Perlman A, Gressel J. Induction of Trichoderma sporulation by nanosecond laser pulses: evidence against cryptochrome cycling. *Photochemistry and photobiology* **51**, 99-104 (1990).
2. Berrocal-Tito G, Sametz-Baron L, Eichenberg K, Horwitz BA, Herrera-Estrella A. Rapid blue light regulation of a Trichoderma harzianum photolyase gene. *Journal of Biological Chemistry* **274**, 14288-14294 (1999).
3. Casas-Flores S, Rios-Momberg M, Bibbins M, Ponce-Noyola P, Herrera-Estrella A. BLR-1 and BLR-2, key regulatory elements of photoconidiation and mycelial growth in Trichoderma atroviride. *Microbiology* **150**, 3561-3569 (2004).
4. Yin T, Wu YI. Guiding lights: recent developments in optogenetic control of biochemical signals. *Pflügers Archiv-European Journal of Physiology* **465**, 397-408 (2013).
5. Zoltowski BD, Vaccaro B, Crane BR. Mechanism-based tuning of a LOV domain photoreceptor. *Nature chemical biology* **5**, 827-834 (2009).
6. Tannous BA, Kim D-E, Fernandez JL, Weissleder R, Breakefield XO. Codon-optimized Gaussia luciferase cDNA for mammalian gene expression in culture and in vivo. *Molecular Therapy* **11**, 435-443 (2005).
7. Diggle SP, Crusz SA, Cámara M. Quorum sensing. *Current Biology* **17**, R907-R910 (2007).
8. Miller MB, Bassler BL. Quorum sensing in bacteria. *Annual Reviews in Microbiology* **55**, 165-199 (2001).
9. Fenno L, Yizhar O, Deisseroth K. The development and application of optogenetics. *Annual review of neuroscience* **34**, 389-412 (2011).
10. Zhao EM, *et al.* Optogenetic regulation of engineered cellular metabolism for microbial chemical production. *Nature* **555**, 683-687 (2018).
11. Motta-Mena LB, *et al.* An optogenetic gene expression system with rapid activation and deactivation kinetics. *Nature chemical biology* **10**, 196-202 (2014).
12. Hallett RA, Zimmerman SP, Yumerefendi H, Bear JE, Kuhlman B. Correlating in vitro and in vivo activities of light-inducible dimers: a cellular optogenetics guide. *ACS synthetic biology* **5**, 53-64 (2016).
13. Tang L. Optogenetic tools light up phase separation. *Nature methods* **16**, 139-139 (2019).
14. Sureda-Vives M, Sarkisyan KS. Bioluminescence-Driven Optogenetics. *Life* **10**, 318 (2020).
15. Kim CK, Cho KF, Kim MW, Ting AY. Luciferase-LOV BRET enables versatile and specific transcriptional readout of cellular protein-protein interactions. *Elife* **8**, e43826 (2019).
16. Berglund K, *et al.* Luminopsins integrate opto-and chemogenetics by using physical and biological light sources for opsin activation. *Proceedings of the National Academy of Sciences* **113**, E358-E367 (2016).
17. Li T, *et al.* A synthetic BRET-based optogenetic device for pulsatile transgene expression enabling glucose homeostasis in mice. *Nature communications* **12**, 1-10 (2021).
18. Prakash M, *et al.* Selective control of synaptically-connected circuit elements by all-optical synapses. *Communications Biology* **5**, 1-13 (2022).
19. Leung SJ, Romanowski M. Light-activated content release from liposomes. *Theranostics* **2**, 1020 (2012).
20. Ghosh S, Carter KA, Lovell JF. Liposomal formulations of photosensitizers. *Biomaterials* **218**, 119341 (2019).
21. WHEELER OH, MATEOS JL. The Ultraviolet Absorption of Isolated Double Bonds¹. *The Journal of Organic Chemistry* **21**, 1110-1112 (1956).
22. Schroeder A, *et al.* Remotely activated protein-producing nanoparticles. *Nano letters* **12**, 2685-2689 (2012).

Reviewers' Comments:

Reviewer #1:

Remarks to the Author:

The authors addressed all my questions and comments. Thank you!